# A Novel Microencapsulated Bovine Recombinant Interferon Tau Formulation for Luteolysis Modulation in Cattle

**DOI:** 10.3390/biom15071009

**Published:** 2025-07-14

**Authors:** Emilio Lamazares, Aleikar Vásquez, Kelly Gancino, Felipe Sandoval, Javiera Yáñez-Torres, Miguel A. Gutierrez-Reinoso, Manuel García-Herreros, Paula Gädicke, Ignacio Cabezas, Florence Hugues, Thelvia I. Ramos, Frank Camacho, Karel Mena-Ulecia, Jorge R. Toledo

**Affiliations:** 1Biotecnology and Biopharmaceuticals Laboratory, Departamento de Fisiopatología, Facultad de Ciencias Biológicas, Universidad de Concepción, Víctor Lamas 1290, P.O. Box 160-C, Concepción 4030000, Chile; elamazares@udec.cl (E.L.); aleikarvasquez@udec.cl (A.V.); felisandoval@udec.cl (F.S.); jyanez2029@udec.cl (J.Y.-T.); 2Grupo de Investigación en Sanidad Animal y Humana (GISAH), Departamento de Ciencias de la Vida y la Agricultura, Universidad de las Fuerzas Armadas ESPE, Sangolquí 171103, Ecuador; kellygancino@gmail.com (K.G.); tiramos@espe.edu.ec (T.I.R.); 3Facultad de Ciencias Agropecuarias y Recursos Naturales, Carrera de Medicina Veterinaria, Universidad Técnica de Cotopaxi (UTC), Latacunga 050150, Ecuador; miguel.gutierrez@utc.edu.ec; 4Instituto Nacional de Investigação Agrária e Veterinária (INIAV), 2005-424 Santarém, Portugal; herrerosgm@gmail.com; 5Pathology and Preventive Medicine Department, School of Veterinary Sciences, Universidad de Concepción, Avenida Vicente Méndez 595, P.O. Box 537, Chillán 3812120, Chile; pgadicke@udec.cl; 6Clinical Science Department, School of Veterinary Sciences, Universidad de Concepción, Avenida Vicente Méndez 595, P.O. Box 537, Chillán 3812120, Chile; oscabeza@udec.cl (I.C.); flohugues@udec.cl (F.H.); 7Recombinant Biopharmaceuticals Laboratory, Departamento de Farmacología, Facultad de Ciencias Biológicas, Universidad de Concepción, Víctor Lamas 1290, P.O. Box 160-C, Concepción 4030000, Chile; fcamacho@udec.cl; 8Departamento de Ciencias Biológicas y Químicas, Facultad de Recursos Naturales, Universidad Católica de Temuco, Ave. Rudecindo Ortega #02950, Temuco 4780000, Chile; kmena@uct.cl

**Keywords:** interferon tau, microencapsulation, luteolysis modulation

## Abstract

Early embryonic loss is a major cause of reproductive inefficiency in cattle, primarily due to premature luteolysis. Interferon tau (IFN-τ), secreted by the trophoblast, plays a critical role in maternal recognition of pregnancy by maintaining corpus luteum function. However, its practical application has been limited by its rapid degradation and short half-life in vivo. Here, we developed a novel formulation of recombinant bovine IFN-τ, combining chitosan-based microencapsulation with starch–chitosan hydrogel delivery, enabling sustained intrauterine release. This dual-delivery strategy offers a significant improvement over conventional IFN-τ administration methods that rely on repeated intrauterine infusions of soluble protein. The rbIFN-τ was expressed in *Pichia pastoris*, purified to 90.1% homogeneity, and structurally validated via homology modeling and molecular docking, confirming its interaction with type I interferon receptors. The encapsulated formulation retained antiviral activity, stimulated transcription of interferon-stimulated genes (*PKR*, *OAS1*, *OAS2*), and showed sustained release in vitro for up to 26 days. In vivo evaluation demonstrated safety and biological efficacy, with treated cattle showing inhibited luteolysis, sustained serum progesterone levels, and preserved corpus luteum integrity. This formulation represents a promising biotechnological approach to improve reproductive efficiency through a long-acting, species-specific IFN-τ delivery system.

## 1. Introduction

Reproductive inefficiency, including early embryonic loss, is a serious cause of economic loss in the global cattle industry. Infertility and embryo loss in dairy and beef herds are estimated to reduce profitability by over USD 3.7 billion annually in the United States [1]. The economic impact of a single pregnancy loss in cattle ranges from USD 90 to USD 1900, driven by reduced calving rates, extended calving intervals, and increased culling rates [2]. In Latin America, where beef and dairy exports are crucial components of the agricultural gross domestic product (GDP), early pregnancy loss in cattle can translate into millions of dollars in direct and indirect yearly losses, particularly in countries like Brazil, Argentina, and Mexico [3]. In Chile, abortion rates overall are underestimated, with reported cumulative incidences around 12.2% and associated economic losses estimated at USD 143 per case [4,5]. These economic losses are compounded in systems that rely on advanced reproductive technologies such as artificial insemination or embryo transfer, where each failed pregnancy represents not only a biological failure but also a wasted technological investment [6,7].

Embryonic loss occurs at a high rate in cattle during the first month [8]. Days 15 and 28 of gestation are crucial because successful gestation maintenance relies on proper maternal recognition of pregnancy [8]. Up to 15% of pregnancies are lost between days 16 and 32 post-fertilization [9]. This failure is primarily associated with insufficient signaling from the embryo, which leads to premature regression of the corpus luteum and, consequently, early disruption of pregnancy maintenance. In ruminants, the signaling molecule responsible for maternal recognition is interferon tau (IFN-τ), a trophoblast-derived type I interferon [10,11,12]. The expression of IFN-τ begins around the development of the blastocoele, when a functional trophectoderm has formed [13,14] and is secreted by mononuclear cells from the embryonic and extraembryonic membranes of ruminants [12]; it inhibits endometrial oxytocin receptor expression and suppresses prostaglandin F2α release [10,11] with a consequent antiluteolytic effect [15,16]. Furthermore, INF-τ has a paracrine effect, inducing interferon-stimulated genes (ISGs) in the endometrium that enable uterine receptivity to the implantation and elongation of the embryo; the ISGs are used for early gestation diagnostics in cows (days 18–20) from circulatory leukocytes [17,18]. The embryo elongation is an essential process for the suitable production of INF-τ and it is controlled by several trophoblastic gene regulators such as DLX3, CDX2, and GATA2/3 [12]. Moreover, INF-τ regulates several genes encoding uterine-derived factors that are essential in the uterus preparation for placenta attachment and early fetal development [12]. IFN-τ secretion peaks between days 15 and 17 and plays a central role in sustaining luteal function and embryo viability [19]. During the estrous cycle, progesterone plays a crucial role by inhibiting the synthesis of prostaglandin F2α (PGF2α) by suppressing estradiol receptors in the endometrium. However, if the oocyte is not fertilized, the endometrium starts to secrete PGF2α. This secretion leads to luteolysis, which causes vasoconstriction and apoptosis of luteal cells, ultimately reducing progesterone production [20]. The decrease in progesterone levels removes the negative feedback on GnRH, initiating a new estrous cycle.

Bovine recombinant IFN-τ (brIFN-τ) has been expressed in various systems, including baculovirus, mammalian, and bacterial platforms [21,22]. However, the methylotrophic yeast (*Pichia pastoris*) offers a superior platform for the cost-effective, scalable production of correctly folded and bioactive rbIFN-τ [23]. Despite efficient production, the complementary application of IFN type I remains limited due to its rapid degradation and short half-life in vivo [24]. To overcome these barriers, controlled release systems based on polymeric micro and nanoparticles have gained attention in veterinary biotechnology. Biodegradable carriers such as chitosan and poly lactic-co-glycolic acid (PLGA) enhance protein stability, prolong systemic residence time and reduce dosing frequency [24,25,26]. Chitosan provides mucoadhesive and pH-responsive properties, while PLGA enables tunable degradation profiles and high drug-loading capacity [27]. Notably, semi-solid hydrogel matrices composed of natural polymers like chitosan and starch could serve as localized delivery platforms for intrauterine administration of encapsulated bioactives [28]. Hydrogels enable sustained release over several days, better mimicking the physiological release pattern of endogenous IFN-τ during early gestation while minimizing systemic degradation and off-target effects [12].

No published studies have explored brIFN-τ encapsulation within polymeric microcarriers and its incorporation into hydrogels to modulate cattle luteolysis. The present study aimed to (i) express and purify biologically active brIFN-τ in *Pichia pastoris*; (ii) formulate and characterize chitosan-based microencapsulation systems; (iii) integrate these into starch–chitosan hydrogels for intrauterine delivery; and (iv) evaluate their biological activity and release kinetics in vitro, safety and anti-luteolytic effects in ruminants.

This biotechnological approach could offer a novel complementary strategy for improving reproductive efficiency in cattle, since it guarantees constant release of IFN-τ as a signal of maternal recognition, particularly when integrated into artificial insemination or embryo transfer protocols. Providing a single-dose sustained-release formulation of brIFN-τ could potentially reduce pregnancy loss rates, lower hormone usage, and increase calving success, ultimately contributing to more sustainable and productive livestock systems.

## 2. Materials and Methods

### 2.1. Modeling by Homology: Computational Procedures

To conduct a homology modeling study, the amino acid sequence of brIFN-τ (database UniProt (G3MZ51·G3MZ51_BOVIN)) in FASTA format was uploaded to the Swiss-Model server [29,30], a web server designed to allow users to build and evaluate protein homology models easily [30,31,32]. The sequence was analyzed inside the server with BLAST algorithm (https://blast.ncbi.nlm.nih.gov/Blast.cgi) accessed on 29 May 2025 [33] and HHBLITS [34] to obtain all available templates. Afterward, the best templates for generating the models were selected based on optimal coverage and percentage of identity (greater than 70%) with known proteins in the Protein Data Bank [35,36].

From the selected templates, the 3D model of the bovine IFN-τ was built by the Swiss-Model web server and validated using the MolProbity algorithm [37] and QMEAN [38]. These parameters tell us if the bovine IFN-τ structure predicted by the homology modeling approach is comparable to what would be expected from experimental structures of similar size and structure [30,39]. To select the best models, the QMEAN parameter was chosen as a reference value from 0 to −4, where a QMEAN value close to 0 suggests a high quality of the homology-modeled protein structure [40]. Structural analysis of the generated models and figures was obtained using the Pymol program Version 2.3.0 [41,42].

### 2.2. Molecular Docking Procedure

To explain the structural basis of the interaction of bovine interferon-tau with its receptors, molecular docking experiments (docking) were performed [43,44,45,46]. For the docking experiments, the structure of brIFN-τ was predicted by a homology modeling approach, while the brIFN-τ receptors Q04790 (α/β-Receptor-1) and F1MXT2 (α/β-Receptor-2) were downloaded from the UniProt database [47], both isolated from the organism Bos taurus (bovine). The receptors were prepared for docking experiments by adding hydrogen atoms to each amino acid at pH = 7.4 (physiological pH), and energy minimization was performed to relax both systems and obtain a local minimum for the structures. This procedure also minimizes the coiled IFN-τ structure predicted by the homology modeling approach using the UCSF ChimeraX program version 1.9 [48,49,50].

Docking experiments were performed with HDOCK [51,52], an algorithm that studies protein–protein interactions. The HDOCK algorithm samples the binding modes between two proteins through a global search method based on the fast Fourier transform (FFT). Then, it evaluates the sampled binding modes with an improved scoring function based on iterative knowledge of protein–protein interactions. The confidence score variable was also considered for selecting the best poses, which is defined by the following expression: (Equation (1))(1)C_s_ = 1.0/(1.0 + e^(0.02×De)+150^)

In Equation (1), C_S_ represents the confidence score, and De represents the docking score calculated by the software program. Following the operation of the HDOCK algorithm, when the confidence score is greater than 0.7, it is very likely that the two molecules will bind stably; when the confidence score is between 0.5 and 0.7, the two molecules could bind, and when the confidence score is less than 0.5 it is unlikely that the molecules will bind. On the other hand, the confidence score should be used with caution due to its empirical nature. From this procedure, 100 docking poses were obtained, and the six best poses for each complex were selected based on the docking score and confidence score for detailed structural analysis.

### 2.3. Expression of Bovine Recombinant IFN-τ in the Pichia Pastoris System

Bovine recombinant IFN-τ expression was performed following the approach described previously [53]. A recombinant clone of *Pichia pastoris* (Mut+, his3) was cultured in five 250 mL flasks containing 50 mL of YP (Yeast Extract 1–Peptone 2%) medium supplemented with glycerol (YPG) (2% *v*/*v*) (PichiaPink™ Expression System, ThermoFisher, Waltham, MA, USA) for 24 h at 30 °C and 200 rpm. Then, the culture was centrifuged at 4200× *g* for 5 min to harvest and resuspend the grown cells in fresh YP medium with methanol at different concentrations (Table 1).

The cultures were maintained at 30 °C, 200 rpm, supplemented with methanol for at least 72 h. The supernatant was obtained by centrifugation at 4200× *g* for 5 min and evaluated by protein electrophoresis (SDS-PAGE) and Western blot analysis using a polyclonal antibody anti-IFN-τ (MyBioSource, Vancouver, BC, Canada). Expression per condition was quantified by densitometry on SDS-PAGE using a bovine serum albumin (BSA) standard curve, as determined by Image StudioTM software 6.0 (Li-Cor Biosciences, Lincoln, NE, USA).

To express brIFN-τ in high-density culture, transformed *Pichia pastoris* was grown in two flasks (2 L) containing 500 mL of YPG medium in each. The cultures were incubated at 30 °C with agitation at 250 rpm for 24 h before being transferred to a 10 L bioreactor (Winpac, Senoia, GA, USA) containing 4 L of MS medium for fed-batch culture [53]. The temperature was maintained at 30 °C, and the pH was regulated at 4.5 by the automated addition of a 25% ammonia solution (NH_3_) and 30% orthophosphoric acid (H_3_PO_4_). Dissolved oxygen (DO) was maintained between 20 and 40%. The agitation speed was 500 rpm in the growth phase and 1000 rpm in the induction phase. In the initial phase, glycerol (2% *v*/*v*) was supplied as the carbon source. Once the DO and pH values indicated that glycerol had been entirely consumed, methanol (0.5% *v*/*v*) was added for the adaptation stage. Upon completion of this stage, the induction of brIFN-τ is initiated by strategy 4 (Table 2).

The inductor addition was automated by a methanol sensor (Raven Biotech Inc., Vancouver, BC, Canada). Traces and vitamins were provided at this stage every 40 h, according to Pedroso-Santana et al., 2020 [53]. The supernatant was recovered by centrifugation (17,000× *g*, 20 min) and immediately stored at −20 °C before further analysis.

### 2.4. Purification of Bovine Recombinant IFN-τ by Ion Exchange

The brIFN-τ purification was performed by sequential cationic and anionic exchange chromatography using an Äkta Prime device (GE Healthcare, Chicago, IL, USA) controlled and monitored by PrimeView software 5.0 (GE Healthcare, Chicago, IL, USA). The supernatant was diluted three times with 50 mM sodium citrate buffer, pH 4.5, and subjected to cationic exchange chromatography with a Giga Cap S 650 s matrix (Toyopearl, Tosoh Bioscience, Minato-ku, Tokyo, Japan). The column was equilibrated with 50 mM sodium citrate buffer, pH 4.5, at a flow rate of 5 mL/min for 8 min. The sample was added at the same flow rate, and elution was performed using a 50 mM Tris-HCl buffer, pH 7.5, at a flow rate of 3 mL/min. The eluted fraction was collected for subsequent purification.

Anionic exchange chromatography was performed on a Giga Cap Q 650 s matrix (Toyopearl, Tosoh Bioscience, Japan). The matrix was equilibrated with 50 mM Tris-HCl buffer, pH 7.5, at a flow rate of 5 mL/min for 8 min. The sample was added at a rate of 3 mL/min. Elution was performed using a saline gradient (1 M NaCl, 50 mM Tris-HCl) at a flow rate of 3 mL/min, and the eluted fractions were collected. The eluted sample was dialyzed on Tris-HCl buffer (50 mM) using a nitrocellulose membrane (12–14 kDa, MWCO) (Fisher Scientific, Waltham, MA, USA). Finally, the sample was concentrated using a three kDa membrane and an Amicon kit (Thermo Fisher Scientific, USA).

### 2.5. Inhibition of the Cytopathic Effect of Mengo Virus in MDBK Cells

The antiviral activity of brIFN-τ was evaluated in Madin–Darby bovine kidney (MDBK) cells affected with the Mengo virus (donation from the Center for Genetic Engineering and Biotechnology, CIGB, Havana, Cuba), using the percentage of cell viability as an indicator of inhibited cytopathic effect. Cells were pre-treated for 24 h with brIFN-τ, followed by 24 h of exposure to viral particles. The tested concentrations were 5.5 ng/L, 1.4 ng/μL, 0.7 ng/μL, 0.3 ng/μL and 0.1 ng/μL of rbIFN-τ. All brIFN-τ treated groups show increased protection against the viral agent, showing higher viability percentages than the negative control (MDBK plus virus without brIFN-τ) but lower than the positive control (MDBK without brIFN-τ and virus). There is also an evident decrease in viability percentage when the amount of brIFN-τ administered to the cells decreases. As a result, the respective percentages of cell viability obtained were: 88.04%, 85.44%, 83.5%, 73.26%, and 64.84%. The negative control reflects 50% viability. The data were processed and plotted in GraphPad Prism 8.0.2 for Windows.

### 2.6. Antiviral Markers Activation Analysis in MDBK Cells

MDBK cells were treated with brIFN-τ at 200, 500, and 1000 ng/mL concentrations for 24 h each. All treatments were performed in triplicate. Then, total RNA was extracted from the cells using the Trizol reagent (Invitrogen, Waltham, MA, USA), following the manufacturer’s instructions. RNA was quantified with the Synergy HTX microplate reader (BioTek Instruments, Winooski, VT, USA). The integrity of the purified RNA was checked using 1% (*w*/*v*) agarose gel electrophoresis in Tris-Acetate buffer solution. For complementary DNA synthesis and real-time PCR, the commercial kit (Thermo Scientific, Waltham, MA, USA) and KAPA SYBR FAST Universal (BIOSYSTEMS, Wilmington, MA, USA), respectively, were used in the AriaMx Real-Time PCR System (Agilent, Santa Clara, CA, USA). The results were analyzed using the comparative Ct (2^−ΔΔCt^) method, and GraphPad Prism 8 software was used to obtain the graphs and statistical analyses. The real-time PCR thermal cycling profile consisted of the reverse transcription step at 50 °C for 30 min. Then, the reverse transcriptase inactivation–initial denaturation step was performed at 95 °C for 10 min, and subsequently, the amplification step consisting of 40 cycles of 15 s at 95 °C and 30 s at a temperature between 56 °C and 60 °C (gene-dependent) for the primers’ annealing step. FA fixed primer concentration of 200 nM was used for the above reaction in a final reaction volume of 12 μL. Specific primers were used to study the relative expression of OAS1, OAS2, PKR, and β-actin genes, which were used as a normalizer (Table 3).

### 2.7. Antiviral Activity of brIFN-τ

MDBK cells were seeded in 96-well plates (Costar, Arlington, VA, USA) at 15 × 10^3^ cells/well in Dulbecco’s Modified Eagle’s Medium (DMEM) + 5% fetal bovine serum (FBS). The plates were incubated at 37 °C and 5% CO_2_ for 24 h. The medium was replaced with 100 µL Mengo virus in DMEM + FBS 2%, making serial dilutions 1:10. The plates were incubated at 37 °C and 5% CO_2_. After 24 h, the plates were washed, fixed, and stained with 50 μL/well of 0.5% crystal violet solution and 20% methanol for 15 min. The plates were washed with H_2_O, and then the crystal violet solution was dissolved with 10% acetic acid (HAc) and 100 µL/well, with agitation at 100 g (xg) (Centrifuge 5702 Eppendorf, Hamburg, Germany) for one hour at room temperature. The absorbance of the plate was read at 590 nm in a Synergy HTX Multi-Mode Reader spectrophotometer (BioTek, Winooski, VT, USA). The data were fitted to a sigmoid curve that determined the mean maximum effective concentration (EC_50_) value, the dilution that generates 50% cell death.

The assay was repeated under the same conditions, exposing the cells to varying interferon concentrations before treatment with the Mengo virus. For commercial standards, the rhIFNα-2b was used (Sigma Aldrich Laboratory, Saint Luis, MO, USA), see [54]. The experimental design consisted of cell control (cc) wells with untreated cells and virus control (cv) cells without interferon treatment exposed to the Mengo virus. The Mengo virus dilution used was the dilution that caused 50% cell death under the same assay conditions. The data were fitted to a sigmoid curve that determined the value of the EC_50_. This dilution generates 50% cell protection and can be defined as the concentration required to obtain a 50% cell protection effect after a specified exposure time. It is also identified with potency [55]. The EC_50_ of a quantal dose–response curve represents the concentration of a compound at which 50% of the population shows a response.

According to the following formula:(2)Abs norm = (Abs-Cv)/(CC-CV)

The data were fitted to a sigmoid curve, and the EC_50_ value was calculated for each case, corresponding to the dilution value that generates 50% cell protection. Taking the IFNα-2b standard as a reference, the interferon titer was calculated from the data provided by the standard and considering the initial concentration, according to the formula:(3)Title rbIFN-τ (IU)/(mL) = (Sample title/Strd title) × Strd (IU/mL)

The specific activity was also determined using the following formula:(4)Specific activity (IU)/(mg) = Title rbIFN-τ (IU/mL)/Concentration (mg/mL)

### 2.8. Encapsulating brIFN-τ Using Chitosan as a Low Molecular Weight Polymer

#### 2.8.1. Microencapsulation of rbIFN-τ in a Chitosan Matrix

The brIFN-τ was expressed in the yeast *Pichia pastoris* and purified, as previously described by our research group (Section 2.3, Section 2.4 and Section 2.5). Microencapsulated low-molecular-weight chitosan was used as a control for incorporation into a semisolid matrix (starch/chitosan, hydrogel). The microencapsulated formulations of the protein brIFN-τ and the control without brIFN-τ were prepared using spray drying technology. Low-molecular-weight chitosan (50–190 kDa, Sigma-Aldrich, St. Louis, MO, USA) was added to deionized water containing 0.5% (*v*/*v*) acetic acid (Sigma-Aldrich, St. Louis, MO, USA) for a final concentration of 8.5 g/L. The mixture was continuously stirred at 22 °C for 16 h at 300 r.p.m.

#### 2.8.2. Preparation of the Samples to Be Microencapsulated

For the empty control (without brIFN-τ protein), 600 mL of 8.5 g/L chitosan solution and 300 mL of 50 mM Tris-base were mixed, and for the sample containing the protein of interest, 600 mL of 8.5 g/L chitosan was mixed with 300 mL of 310 µg/mL brIFN-τ solution. Spray drying was performed in the mini spray-dryer B-290 (Büchi, Flawil, St. Gallen, Switzerland) [56]. Three drying sequences were conducted, testing different temperatures and keeping the rest of the parameter’s constant (Table 4).

The resultant suspension was fed into the nozzle with a diameter of 0.7 mm. The microparticles were collected and stored at −20 °C, and the empty microparticles were used as the negative control. The percentage of yield obtained in the microencapsulation process by spray drying was calculated using the following formula:Yield (%) = [Mass obtained (g)/(Sample volume (L) × 8.5 g/L)] × 100 where 8.5 g/L corresponds to the total mass of solids used in the encapsulation.

The morphological characterization and size determination of the microparticles obtained were performed by scanning electron microscopy (SEM).

### 2.9. Microencapsulated brIFN-τ Release Assay Generated at Three Temperatures

Using microparticles generated at three temperatures (100 °C, 120 °C, and 140 °C), brIFN-τ time–release assays were performed. For this, 10 mg of encapsulated brIFN-τ was resuspended in 300 μL citrate buffer (50 mM, pH 6.8). These resuspensions were subjected to slow stirring (100 rpm) at 37 °C for 48 h, and 100 μL of the respective samples were withdrawn at the indicated times. SDS-PAGE and Western blot determined the brIFN-τ release at a fixed time.

### 2.10. Release Assay of brIFN-τ from Chitosan Microparticle from 120 °C

An interferon release assay was conducted using chitosan microparticles produced at 120 °C, simulating conditions in the bovine uterus. In a single test tube, 20 mg of microparticles containing IFN-β were dissolved in 300 µL of 50 mM citrate buffer at pH 6.8. The mixture was then incubated at 37 °C with continuous agitation at 100 r.p.m. Samples were collected on day 0, 1, 5, 10, 15, 20, 21, 22, 23, 24, 25, and 26 of the assays. Each sample was centrifuged at 10,000 r.p.m. for 5 min. A measure of 100 µL of the supernatant was extracted from each sample, and the same volume of citrate buffer was added to maintain the original condition. Additionally, the release of brIFN-τ was evaluated by quantifying the total protein in each extracted supernatant using microBSA (micro BCATM Protein Assay Kit, Thermo Scientific, Waltham, MA, USA). Finally, SDS-PAGE and Western blotting techniques were employed for further analysis. Original figures can be found in Appendix A.

### 2.11. Incorporation of Microencapsulated brIFN-τ into a Semi-Solid Matrix (Starch/Chitosan, Hydrogel)

Semi-solid matrices were synthesized by chemical cross-linking with genipin. A 1% chitosan solution (0.5% acetic acid, pH 4.5) and an 8% starch solution (heated at 120 °C for 15 min) were prepared. Both solutions were mixed in a 30% chitosan/70% starch ratio and kept under 300 r.p.m. agitation for 24 h at 25 °C. Genipin was added to a final concentration of 0.05% (*v*/*v*), and immediately, the microparticles loaded with brIFN-τ 0.5 µg/mL and the empty microparticles were added. The mixture was kept at 300 r.p.m. agitation for 5 days at 25 °C. The hydrogels were transferred to the final molds and subjected to three cycles of freezing (−20 °C for 12 h)–thawing (25 °C). Finally, they were frozen at −80 °C in an ultrafreezer for 2 h and freeze-dried for 24 h. The semi-solid matrices obtained were stored at room temperature until use.

### 2.12. The Release Assay of Microencapsulated brIFN-τ Associated with Hydrogel

In this case, 6-well plates, maintaining conditions such as the release of brIFN-τ from microencapsulated hydrogels consisting of microencapsulated plus brIFN-τ, soaked in 10 mM citrate buffer pH 6.5 and hermetically sealed to prevent water loss by evaporation, were kept under slow stirring conditions at 37 °C for 26 days. In this case, 1 mL of each sample was removed at the established times, and immediately, the same volume was replaced with 50 mM citrate buffer (pH 6.8).

### 2.13. Safety Assessment of Intrauterine Device in Ovine Model

This trial used 10 clinically healthy and non-pregnant ewes (age: 6 mo.). The animals were acclimated for 15 days before the beginning of the experiment. During this time, to evaluate the individual manifestation of estrus, two synchronization protocols, separated for 7 days, were established, based on the administration of prostaglandin (dinoprost 0,5%, Lutalyse™, Zoetis, Santiago de Chile, Las Condes, Chile). The protocol begins with administering 5 mg/ewe of intramuscular prostaglandin at time 0. After seven days, a second dose of prostaglandin is given. Twenty-four hours later, when estrus is detected, the implant is applied. This procedure ensures that estrus occurs when administering the intrauterine brIFN-τ implant, which typically coincides with mating or insemination. By doing this, the cervix remains open, allowing for easier insertion of the injector to place the implant in the uterus. After this period, the animals were randomly assigned to two groups of five each. Group 1 (G1) received the intrauterine brIFN-τ implant, while group 2 (G2) was left as a control. First, both groups were synchronized with the same prostaglandin protocol. Twenty-four hours after the synchronization and with evident estrus manifestation, the respective treatments were applied. For group 1, each animal was first desensitized with 1 mL of lidocaine (2%) in a low epidural injection (Drag Pharma^®^, Santiago de Chile, Chile). Then, each ewe was restrained on its back on a gynecological table, and the interferon was administered in the uterine horn with a semen straw injector of 4 mm diameter and, with the help of a speculum, through the cervix. Each animal in both groups was evaluated according to the animal supervision protocol. During the first week, the evaluation was carried out every 24 h between 08.00 and 08.15 a.m. and later on days 12, 18, and 24. This protocol considers monitoring spontaneous behavior and responding to manipulation, pain, aspects of the fur, secretions, water, and food consumption, assigning a point score from 0 to 3 to each item [56,57]. Furthermore, to establish possible local effects of the interferon, rectal temperature (digital thermometer, ecomed^®^, mod. TM-60E, Medisana^®^, Neuss, Bundesland, Deutschland) was registered at the exact checking times, as well as the condition of the genitalia: redness, local pain (response to pressure), edema, secretions and vulvar temperature (infrared thermometer, mod. JPD-FR202, Shenzhen Jumper Medical Equipment Co., Shenzhen, Guangdong, China). Also, to assess possible variations in blood count, 6 mL samples were taken with the Vacutainer^®^ system from the jugular vein on days 0, 3, 7, and 14.

### 2.14. Anti-Luteolytic Activity in Cows

Progesterone levels were evaluated using 30 healthy, non-pregnant, hybrid beef cattle cows. The animals were randomly assigned to three groups: group 1, 10 cows treated with microparticles plus hydrogel without brIFN-τ (control group); group 2, 10 cows treated with brIFN-τ microencapsulated brIFN-τ plus hydrogel (brIFN-τ micro/hydrogel); group 3, 10 cows treated with brIFN-τ non-encapsulated brIFN-τ plus hydrogel (brIFN-τ hydrogel). All the cows were synchronized with a standard protocol [58] based on the application on day 0 of an intravaginal progesterone implant (Easy Breed^®^, Zoetis, Parsippany, NJ, USA) and 2 mL of estradiol benzoate (Syntex™, Palo Alto, CA, USA) and day 8, extraction of the implant and administration of 0.5 mL of estradiol cypionate (E.C.P., Zoetis, Chile) + 5 mL prostaglandin (Lutalyse™, Zoetis, Chile).

Between 52 and 56 h after the end of the protocol, all the animals were implanted with the respective semisolid hydrogel trans cervically, using a semen straw injector of 4 mm diameter. Immediately after the implantation, the implant’s presence in the uterus was assessed by transrectal ultrasonography examination (Mindray DP50 Vet). Ultrasonographic monitoring in the cows was performed seven times between days 0 and 22 to determine the disintegration rate of the implant. Moreover, during the first three days after the implant’s administration, each animal’s rectal temperature was recorded to assess the safety of the treatments. After the synchronization protocol and implant, the serological progesterone levels in the animals were evaluated for 24 days to determine the progesterone curve concerning the estrus cycle and the anti-luteolytic effect. Blood samples were taken in a 9 mL tube without anticoagulant, with a Vacutainer system from the jugular vein on days 0, 7, 15, 19, and 24. The samples were centrifuged at 3500 r.p.m. for 10 min. The serum was frozen at −20 °C in 1.5 mL Eppendorf tubes until further processing. Finally, the samples were analyzed by radioimmunoassay (RIA) in an external laboratory.

### 2.15. Statistical Analysis

The statistical analysis was conducted in three sequential steps. First, descriptive statistics were computed to summarize the data distribution. Categorical variables were expressed as frequencies and percentages, while continuous variables were evaluated for normality using the Shapiro–Wilk test and visual inspection of Q-Q plots. Normally distributed variables were reported as mean ± standard deviation (SD), whereas non-normally distributed variables were presented as median and interquartile range (IQR). Second, the assumptions of parametric tests were verified by assessing normality (Shapiro–Wilk test) and homogeneity of variance (Levene’s test for parametric analyses or Bartlett’s test for non-parametric comparisons). Finally, hypothesis testing was performed based on data characteristics: categorical data were analyzed using Pearson’s chi-square test (or Fisher’s exact test for minor, expected frequencies), while continuous variables were compared using Student’s *t*-test (for normally distributed data) or Mann–Whitney U test (for non-normal distributions). For multiple group comparisons, one-way ANOVA (with Tukey’s post hoc test) or Kruskal–Wallis test (with Dunn’s post hoc correction) was applied as appropriate. All tests were two-tailed, with statistical significance set at *p* < 0.05. The complete analysis was performed using GraphPad Prism 5, ensuring reproducibility of the results.

## 3. Results

### 3.1. Computational Modeling of the 3D Structure of the Interferon Tau

#### 3.1.1. Homology Modeling

The starting point for homology modeling is reported in Figure 1A, where the amino acid sequence of bovine IFN-τ is shown, as previously defined from the database UniProt (G3MZ51 · G3MZ51_BOVIN) [47].

Based on this sequence, a comprehensive search for suitable structural templates was conducted to construct homology-based 3D models. A total of 25 candidate templates were evaluated (Table 5).

The structure with a QMEAN value closest to 0 and the lowest RMSD, 453, was selected among the generated models. These criteria indicated high structural accuracy and consistency with the reference template. The chosen model accurately represents the three-dimensional architecture of bovine IFN-τ (Figure 1B). The model displays structural features consistent with those of functional interferon, supporting its suitability for further structural and interaction analyses.

#### 3.1.2. Docking Assays

The top five binding poses, along with their corresponding docking scores and confidence scores for both receptor complexes, are reported in Table 6.

The conformations had a docking score below -200 kcal/mol, indicating that the complexes between bovine IFN-τ and α/β-Receptors-1 and 2 are stable (Table 5). However, the most negative docking score was found in the complexes formed with IFN-τ and α/β-Receptor-2, indicating a higher affinity between these two proteins. The docking experiments show that the interaction between IFN-τ and its type-1 receptor generated seven hydrogen bond interactions, only three smaller than 3 Å (Figure 2).

These interactions are also strong and capable of stabilizing a protein–ligand complex. In this complex (IFN-τ-α/β-Receptor-1), the strongest interactions were found between the carbonyl oxygen of Gln65 of IFN-τ with the NH2 of Asn172 of the receptor at 2.0 Å and the NH2 of the side chain of Arg47 of IFN-τ with the carboxyl group of the side chain of Asp39 of the receptor at 2.1 Å. Another strong interaction was found between the carboxyl group of Val68 of IFN-τ and the NH2 group of Asn222 of the receptor at 2.6 Å (Figure 2A). The strongest interactions were found between the amino group of Gln45 of IFN-τ with the carboxyl terminal of Glu216 of the receptor at 2.1 Å and the NH2 group of the functional group of Arg47 of IFN-τ with the carboxyl group of Lys218 of the receptor at 2.8 Å (Figure 2B). The other interaction considered strong in this complex is the other nitrogen of the amino group of Gln45 of interferon with the carboxyl terminal of Glu216 at 3.1 Å. These two hydrogen bonds formed by Gln45 of interferon confer a high stability to this complex.

### 3.2. Expression, Characterization, and Purification of brIFN-τ

#### 3.2.1. brIFN-τ Expression

SDS-PAGE analysis of the culture supernatants revealed two protein bands of approximately 17 kDa and 20 kDa, absent in the negative control. Western blot analysis confirmed the recognition of both bands by anti-bovine IFN- antibodies, suggesting the presence of two distinct isoforms of brIFN-τ (Figure 3).

The protein expression after 72 h of induction varied significantly depending on the induction strategy employed: 41 μg/mL for strategy 1, 55.54 μg/mL for strategy 2, 58.55 μg/mL for strategy 3, 72.28 μg/mL for strategy 4, and 30.03 μg/mL for strategy 5. At 120 h (Figure 3C,D), the measured protein yield was 97.35 μg/mL for strategy 3 and 105.05 μg/mL for strategy 4 (Figure 3E). This latter consistently yielded a higher brIFN-τ concentration than strategy 3 across the time course (Figure 3E, blue line vs. red line).

#### 3.2.2. Purification of brIFN-τ

The culture supernatant obtained from strategy 4 was subjected to a chromatographic purification process. The first step showed a protein fraction with 80.1% purity and the elution profile a single major peak corresponding to the target protein (Figure 4A, E1).

The second step generated three distinct elution peaks (Figure 4B, E1, E2, and E3), with brIFN-τ present only in the first two elution peaks (Figure 4C). The final purity reached 90.1%, and the overall recovery yield of brIFN-τ was 45.63%.

#### 3.2.3. Cell Viability, Antiviral Markers, and Antiviral Activity of brIFN-τ

The cell viability assay showed that purified brIFN-τ maintained 100% cell survival up to a 1:128 dilution after 24 h of exposure, indicating complete protection against virus-induced cytopathic effects. In contrast, human IFNα-2b (hIFNα-2b), used as a positive control, reduced cell survival starting at the 1:32 dilution, with lower viability observed at higher dilutions (Figure 5, orange bars).

Both brIFN-τ and hIFNα-2b treatments resulted in significantly higher cell viability (*p* < 0.05) compared to the negative control. The cell viability assays performed in MDBK cells exposed to the Mengo virus confirmed the consistent protective activity of brIFN-τ over hIFNα-2b across all tested dilutions (Figure 5C). The specific antiviral activity of brIFN-τ was approximately 9.7 × 10^9^ IU/mg (Figure 5D). No statistically significant differences were observed between the brIFN-τ doses tested, whereas all treated groups showed significantly higher expression (*p* < 0.05) than the untreated control group.

### 3.3. Microencapsulation of brIFN-τ and Packaging in Semi-Solid Matrix

#### 3.3.1. Microparticles Obtained by Spray Drying Procedure

During the microencapsulation procedure of brIFN-τ (Section 2.9), the total yield exceeded 56% (Table 7).

The results from spray drying the empty chitosan control samples (without brIFN-τ) indicate that the encapsulation yields at different temperatures are similar (Table 8).

Shape and size of the empty chitosan particles and the chitosan particles encapsulating brIFN-τ are reported in Figure 6A–F.

The control empty chitosan particles were regular, irrespective of the temperature conditions, and remained relatively consistent (Figure 6A,C,E). In contrast, adding brIFN-τ resulted in more irregular particle morphology and increased surface heterogeneity, attributed to the presence of the recombinant protein (Figure 6B,D,F).

#### 3.3.2. Antiviral Activity of Encapsulated brIFN-τ In Vitro

Significant differences (*p* < 0.05) in cell survival were observed between treated groups and the negative control. Western blot analysis (Figure 6H) showed that brIFN-τ was present in supernatants collected after 48 h of incubation, with higher released levels (*p* < 0.05) observed for particles generated at 140 °C and 120 °C, relative to those at 100 °C (Figure 6H, lanes 1 to 3).

#### 3.3.3. brIFN-τ Release In Vitro, Simulating Bovine Uterine Environment Conditions

The results indicate that brIFN-τ can be detected in the supernatant as early as day 1, with concentrations reaching approximately 100 μg/mL (Figure 7A).

However, the levels subsequently declined until day 22, when the signal was observed in Western blot analysis (Figure 7B).

### 3.4. Semi-Solid Starch/Chitosan Matrix with Incorporated Microencapsulated brIFN-τ

#### 3.4.1. Stable Hydrogel Matrix Synthesis

The starch-to-chitosan ratio was set at 70:30 to balance stiffness, flexibility, and long-term stability (results). Following the addition of the crosslinking agent, microparticles were introduced on day 2 of the procedure (Figure 8). At this stage, no structural loss was observed in the hydrogel by the end of the process (Figure 9A).

#### 3.4.2. Hydrogel Characterization

The SEM results indicated that the hydrogel created with microparticles containing brIFN-τ (Figure 9A) exhibited greater structural homogeneity than those produced with soluble brIFN-τ and empty microparticles. Case A demonstrated less heterogeneity despite all samples showing cavities within the 100 to 200 μm range (Figure 9). In contrast, the hydrogels formed under the alternative conditions displayed greater variability in the cavities observed (Figure 9B,C). The results showed that the hydrogels containing microparticles with 730 brIFN-τ provided 100% protection up to a dilution of 1:8. However, the protection began to decrease with a dilution of 1:16, reaching non-significant values at 1:32 (Figure 9, white bars). In contrast, the hydrogels with unencapsulated soluble brIFN-τ exhibited lower protection, around 70% up to a 1:4 dilution, with subsequent decreases observed. For samples containing only purified brIFN-τ, protection was maintained up to 1:128 dilution (Figure 9D, black bars). Significant differences (two-way ANOVA *p* < 0.05; Shapiro–Wilk *p* > 0.10; Levene’s *p* > 0.10) were noted between the black and white columns compared to control cells treated with buffers and those treated with supernatant from hydrogels containing empty particles (*p* < 0.05).

#### 3.4.3. brIFN-τ Release from Hydrogel In Vitro

Total protein values (Figure 9E) were first detected after 24 h, showing a steady increase until the tenth day, when the maximum release was recorded at 22.7 ± 1.39 μg/mL. After this peak, the protein release remained relatively constant until the end of the test on day 26, with values ranging from 17.8 ± 1.76 to 20.48 ± 2.36 μg/mL. The cumulative amount of protein released over the 26-day evaluation reached 214.4 μg/mL. Throughout the study, from day 5 to day 20, an 80% level of protection was measured, which remained above 70% until day 25 (Figure 9F) (two-way ANOVA *p* < 0.05 Shapiro–Wilk *p* > 0.10; Levene’s *p* > 0.10).

### 3.5. Drug Safety and Anti-Luteolytic Activity

#### 3.5.1. Device Safety in the Ovine Model

Figure 10A reports the protocol used for drug safety.

Regarding the behavioral assessment of sheep during the trial, no behavioral changes were detected in either group. The total individual scores for animals in both groups remained between 0 and 3, indicating that the sheep maintained normal, stable conditions and that the formulation did not impact on their well-being. Concerning examination of the genital area and secretions, all sheep exhibited redness, edema, and mucous secretion on day 0, which was associated with the onset of estrus (Figure 10B,C). During the first week following treatment, there was no increase in these indicators, nor were any secretions present. Only variations in redness, edema, and secretions characteristic of the estrous cycle 800 were observed throughout the trial, with no significant differences identified between the trial groups. Pairwise comparisons indicated that on days 7 and 24, the control group had significantly lower rectal temperatures than the treated group (*p* < 0.05); however, these differences were not considered clinically relevant. Some measurements below 38 °C were noted when ambient temperatures fluctuated between 2 and 3 °C (Figure 10D). Vulvar temperature data were analyzed using the same statistical approach. The assessment of vulvar temperature showed an average of 36.7 °C for the treated group and 37.0 °C for the control group at the start of the trial, when all females were in heat. Following the application of treatment, the average vulvar temperature fluctuated between 36.7 °C and 37.9 °C for both groups, with overall averages of 37.4 °C for the treated group and 37.2 °C for the control group, revealing no significant differences between them (*p* > 0.05) (Figure 10E). The hematology analysis showed that values were within normal ranges for both groups at all assessed time points (Table 9).

With regard to total leukocytes, the control group exhibited abnormal values on day 7 but returned to normal levels on day 14. Despite these values being outside the normal range, there were no significant differences (*p* > 0.05) between the treated and control groups.

#### 3.5.2. Anti-Luteolytic Activity Validation in Cows

Figure 11A reports the protocol used for anti-luteolytic activity validation.

Ultrasound data revealed that the administered device remained visible until day 22 in the group with brIFN-τ microencapsulated with hydrogel (G2), whereas the non-encapsulated brIFN-τ plus hydrogel group (G3) did not show visibility of the device until day 11 (Figure 11B). Progesterone concentration underscores the hormonal kinetics during the test, demonstrating higher and more sustained levels (*p* < 0.05) throughout the estrous cycle in G2 compared to the control group and notably maintaining higher levels (*p* < 0.05) than those observed in G3 (Figure 11D). The results indicated decreased circulating progesterone levels in the control group due to the luteolytic effect (characteristic of a normal estrous cycle). Conversely, this decline was not observed in the group receiving microencapsulated brIFN-τ (G2) and only to a lesser extent (*p* < 0.05) in the group with non-encapsulated interferon (G3) (Figure 11D).

## 4. Discussion

### 4.1. Homology Modeling of Bovine Recombinant IFN-τ

Homology modeling is an essential computational tool for predicting the three-dimensional structure of a protein when its crystallographic structure is unavailable [58,61]. Based on the bovine IFN-τ sequence in the existing literature, we searched for potential templates and selected the protein with the UniProt database code AOA7R8C394.1. A [47] to build the bovine IFN-τ model. This protein was chosen as the template due to its 100% sequence identity with bovine IFN-τ, indicating it as a closely related homolog with a high coverage of 98%, suitable for model construction (Figure 1A). The model was constructed using the SwissModel server [29,30]. We selected the best models based on QMEAN [45] values closest to 0 and lowest RMSD [38,62] (Figure 1B). In this context, the chosen model effectively represents the folding of bovine IFN-τ, which is expected to resemble that of other type I interferons (alpha, beta, and omega) and exhibits a predominance of alpha helices over beta helices (Figure 1C,D). This finding is consistent with existing literature [63]. Based on these results, we can hypothesize that the structure of bovine IFN-τ obtained through homology modeling will benefit further studies, such as analyzing interferon point mutations and their interactions with receptors. However, it may have limitations in loop regions, where homology modeling tends to be less accurate.

### 4.2. Docking Experiment of IFN-τ with the α/β-Receptor 1 and 2

Based on homology modeling, docking simulations were conducted to predict the binding modes of bovine IFN-τ with its receptors, α/β-Receptor 1 and 2. The results show stable interactions, with a greater affinity observed for α/β-Receptor 2 (Figure 2B). All poses from the docking experiments exhibited interaction energies below −200 kcal/mol, indicating a high probability of stable binding between interferon and the receptors (Figure 2). The most favorable binding energy was found in the complex formed by IFN-τ and α/β-Receptor 2, with a value of −271.82 kcal/mol for Pose 1. This result suggests a powerful affinity of IFN-τ for α/β-Receptor 2 (Figure 2B).

The confidence scores [51,64] obtained for this complex support the reliability of the predictions, with values ranging from 0.84 to 0.92. The stability of the IFN-τ-α/β-Receptor 2 complex is attributed to the hydrogen bonding interactions observed (Figure 2, dashed yellow line). Specifically, interactions between Gln45 and Arg47 of interferon with Glu216 and Lys218 of receptor 2 had distances of less than 3 Å, significantly contributing to the stability of the IFN-τ-α/β-Receptor 2 complex (Figure 2B).

The dual interaction of Gln45 in IFN-τ is particularly noteworthy, as it appears to play a crucial role in stabilization. The enhanced interaction of IFN-τ with α/β-Receptor 2 aligns with known mechanisms of interferon signaling, suggesting that differential receptor binding may influence the activation of the JAK-STAT signaling pathway [52,65,66].

### 4.3. Expression, Purification, and Biological Activity Evaluation of brIFN-τ

The AOX1 promoter sequence was utilized to produce brIFN-τ, leveraging the methanol utilization capability of *Pichia pastoris* [67]. The AOX1 promoter is a strong promoter activated by methanol and inhibited by glucose and ethanol [68]. During the induction phase, the protein expression level is influenced by the concentration of methanol provided [69]. Consequently, several induction strategies based on different methanol concentrations were evaluated to identify the optimal method for brIFN-τ expression (Table 1, strategies 1–5). Experiments conducted in shaken flasks containing 50 mL of culture demonstrated that optimal brIFN-τ expression occurred with an initial methanol concentration of 0.5% (Figure 3A). This concentration was later increased to 1%, accompanied by an extended incubation period of 120 h (strategy 4) (Figure 3D). Previous studies have shown that the best yields of heterologous protein expression in fed-batch fermentation are achieved with 0.5% *v*/*v* methanol supplementation over 72 h [70]. In contrast, Zhang et al. (2009) indicated that 1.5% methanol is the most effective concentration [71]. Strategy 4 is a procedure previously used to express various recombinant proteins using *P. pastoris* in a low-scale culture [68,72]. A correlation between the increase in optical density (OD) and the accumulation of brIFN-τ in the culture supernatant was observed, particularly when comparing strategy 4 to strategy 3 (Figure 3E). Our findings align with those of Li et al. (2013), who indicated that a 1% methanol concentration is optimal for inducing recombinant lipase expression in *Pichia pastoris* cultures grown in shake flasks [73]. They noted that higher levels of methanol reduced enzyme activity and protein yield, likely due to cellular toxicity. The results from our SDS-PAGE and Western blot analyses revealed the presence of two bands (~17 kDa and ~20 kDa) (Figure 3 and Figure 4C), consistent with earlier reports on recombinant interferon expression using the *Pichia pastoris* system. For instance, Pedroso-Santana et al. (2020) identified a similar band range for porcine IFN-α, suggesting various conformations [53]. Meanwhile, He et al. (2019) attributed this observation to glycosylation [74]. Given that our protein has a potential glycosylation site, it is reasonable to conclude that the lower band corresponds to the non-glycosylated form of brIFN-τ. In contrast, the upper band represents its glycosylated counterpart.

The purification process benefited from the low level of endogenous protein secretion in *Pichia pastoris*, which is a significant characteristic of this expression system [75]. Other interferons expressed in *P. pastoris* were successfully purified using ion exchange chromatography, achieving high purity levels. For example, Wang et al. (2024) reported a purity of 96.7% in a single step of cation exchange chromatography [75]. Our study estimated purity after cation exchange chromatography was 80.1% (Figure 4A,C,D). However, this figure should be confirmed using a more accurate method, such as high-performance liquid chromatography (HPLC). Nagaya et al. (2004) described a more complex purification process that achieved 91% brIFN-τ purity in a baculovirus system [21]. This result highlights a significant advantage of using *Pichia pastoris* for brIFN-τ production compared to the baculovirus system. In our case, after two chromatographic steps, we attained a purity of 90.1% for brIFN-τ in the *P. pastoris* expression system (Figure 4B,C), which is close to what was previously reported [21].

The biological activity of brIFN-τ was assessed by evaluating its ability to inhibit the cytopathic effect induced by the Mengo virus in MDBK cell cultures derived from bovine kidney epithelial cells [76]. There is currently no commercial IFN-τ available, so the results were compared to those of a commercial human standard. Analysis of the brIFN-τ results indicated 100% survival of cells treated with dilutions up to 1:128 after 24 h of exposure, demonstrating complete protection (Figure 5A, black columns). A similar response was observed with human IFNα-2b used as a control (Figure 5A, blue columns). However, the survival percentage decreased starting from a dilution of 1:32. This trend can be explained by the species-specific nature of interferons [77], which means their activity can vary between species. This is expected since the MDBK cell line originates from bovine tissues, not human cells.

Significant differences were noted between brIFN-τ and human IFNα-2b effects compared to the cells treated only with the buffer (negative controls). These findings support evidence that brIFN-τ has biological activity. Our results are aligned with previously published studies that demonstrated the activity of interferons expressed in *P. pastoris* systems [78,79].

The specific activity values obtained for brIFN-τ exceed those reported by Li and Roberts (1994), who generated a series of ovine IFN-τ variants prepared with alterations in the carboxyl-terminal region and showing maximum values of 9.3 × 10^7^ IU/mg in a double mutant termed (S1 + C: ThràIle) without significant differences, concerning the wild type (7.1 × 10^7^ IU/mg), both under specific activity [80]. These differences may be based mainly on the expression systems used in each case.

To confirm the functionality of brIFN-τ, in addition to considering cell survival, IFN-stimulated genes (ISGs) were evaluated. Since interferons (IFNs) induce the overexpression of several antiviral agents, including 2′,5′-oligoadenylate synthetase (OAS1 to OAS3) and protein kinase R (PKR) [81,82], which inhibit viral replication in infected cells [78], an elevated expression of OAS1, OAS2, and PKR can be considered a hallmark of viral infection and IFN stimulation. Their expression and activation reflect the cell’s attempt to inhibit viral replication and spread. As ISGs, they may be used as molecular markers to evaluate the presence and strength of antiviral responses in both research and clinical settings. Their utility as biomarkers and their mechanistic roles in RNA degradation and the inhibition of translation highlight their importance in host–pathogen interactions and immune regulation [80,83]. The increased expression of OAS1, OAS2, and PKR transcripts observed when MDBK cells were exposed to brIFN-τ, compared to untreated cells, is clear evidence of both the recombinant protein functionality as well as its cellular protection against Mengo virus (Figure 5D). The relative values or fold-changes of the evaluated genes are within the range seen in experiments where these genes were assessed following exposure of cells to type I IFN [84].

### 4.4. Microencapsulation of brIFN-τ

The encapsulation of different bioactive components is essential for protecting these molecules from adverse storage processes and conditions, such as high temperatures, high humidity, high oxygen levels, specific pH values, and light exposure [85]. Furthermore, this technology can produce customized ingredients, additives, and supplements with extended shelf lives that can be applied to food, pharmaceutical, and cosmetic products [86]. Another advantage of encapsulation is the controlled release of bioactive ingredients in commercial products or biological systems [24]. Spray drying is one of the most popular, simple, economical, and common encapsulation technologies. This process is very short (seconds), so even heat-sensitive ingredients can be encapsulated by spray drying. It has been used for nutraceuticals, probiotics, proteins, and peptides [85]. Our main objective was to protect interferon and increase its availability over time. In the results described, different encapsulation temperatures were analyzed to establish the optimal working parameters based on the yield and morphology of the obtained particles. Considering the encapsulation yield between the three temperatures studied, higher yields were achieved at 120 and 140 °C than at 100 °C. If we analyze the diameter of the microparticles obtained, regardless of the temperature, no differences were observed between the particles (Figure 6B–F). In the in vitro viability analysis, the encapsulated interferon in the three conditions showed 100% protection up to 1:16 dilutions; from 1:32, a decrease was observed in all three conditions (Figure 6G). In the case of the microparticles generated at 100 °C, approximately 80% protection was observed for 1:32 and 70% at 1:64. This may be justified because the microparticles were generated at the lowest temperature, thereby preserving more active ingredients than the others. The outcome is influenced by the active ingredient employed. The temperature range for the microencapsulation of proteins and thermolabile compounds in biomedical research is between 60 °C and 150 °C to prevent protein denaturation. For instance, temperatures between 120 °C and 150 °C are recommended to microencapsulate probiotics and recombinant proteins [87]. Analyzing the brIFN-τ release from the microparticles, practically all the brIFN-τ had been released after 48 h from the microparticles obtained at 140 °C, less released at 120 °C, and hardly any was released from the 100 °C microparticles (Figure 6H, lanes 1–3). Considering all the factors, it was decided to use the microencapsulation method at 120 °C based on the conservation of biological activity, adequate size of microparticles between 3 and 15 μm [88], active ingredient partial release, and better performance compared to that obtained at 100 °C.

The time–release of brIFN-τ from microparticles generated at 120 °C was assessed in vitro, simulating conditions in a cow’s uterus. The release was observed up to day 20 of the test, confirming that it was sustained over time (Figure 7). These results are consistent with those reported by Fleitas-Salazar et al. (2022), who encapsulated IFN-α using an emulsion and spray-drying method with PLGA and chitosan [59]. In their study, spray drying was performed at a temperature of 95 °C, and the release of IFN-α extended up to 26 days, with sustained bioactivity for at least 15 days. In contrast, our formulation, produced at a higher temperature (120 °C), showed a sustained release of brIFN-τ up to day 20, with a gradual release profile. Furthermore, brIFN-τ was detected using Western blot analysis, which indicated a peak release between days 1 and 5, gradually decreasing until day 20. These results demonstrate that the microparticle contents can be released over time, lasting until day 20 of the test. Considering that the average length of a cow’s estrous cycle is 19–21 days, with drops in progesterone levels observed from days 14–16 [89], the microencapsulated brIFN-τ could help maintain anti-luteolytic conditions in the animal until at least day 19 of the cycle. This fact suggests that treatment with microencapsulated brIFN-τ could extend the cycle in the treated animals.

### 4.5. Semi-Solid Matrix brIFN-τ Incorporation

The present study tested various starch–hydrogel formulations to incorporate the interferon-containing microparticles into a semi-solid matrix that allows administration using the same cannula as in artificial insemination processes. We focused particularly on their ability to absorb samples and their rigidity versus flexibility, which enables a slower release of the active ingredient while leveraging the inert properties of the compound [90]. We tested different ratios of 8% starch and 1% chitosan during the study. Ultimately, we chose a combination of 8% starch and 1% chitosan in a 70/30 ratio. This formulation provided an optimal balance of hardness and flexibility compared to other tested ratios. In contrast, hydrogels made solely with 8% starch produced a very rigid structure prone to breakage [91], making them unsuitable for animal intrauterine application devices.

After obtaining the hydrogels (Figure 8), their characteristics were analyzed using scanning electron microscopy (SEM) (Figure 9A–C). The hydrogel containing microencapsulated brIFN-τ exhibited lower heterogeneity than those without this combination (Figure 9A vs. Figure 9B,C). The structural difference in the photomicrographs of the three hydrogel samples may be attributed to adding chitosan microparticles with brIFN-τ. This addition introduces charges to the mixture during crosslinking, likely promoting a more homogeneous hydrogel structure [92]. In contrast, greater heterogeneity was observed in the other samples. In one case, this could be due to the addition of a soluble brIFN-τ, which, despite being a small volume, can significantly affect the mixture (Figure 9B). Additionally, when empty chitosan microparticles were added, no other fillers were included, which may account for the heterogeneity seen in the control hydrogel (Figure 9C).

In our cell viability analysis using the supernatants released from each generated hydrogel, we found that the hydrogels with microencapsulated brIFN-τ maintained 100% protection even at 1:32 dilution (Figure 9D, white columns). In contrast, the hydrogels with soluble brIFN-τ that were not encapsulated (Figure 9D, gray columns) showed less than 80% protection starting from a dilution of 1:4. This discrepancy may be attributed to the lack of protective effects from soluble brIFN-τ before its incorporation into the hydrogel production process.

As expected, the cells treated with soluble interferon, whether encapsulated in the hydrogel or not, maintained protection levels up to 1:128, like the results previously obtained (Figure 5A, black columns).

### 4.6. Safety and Anti-Luteolytic Effects In Vivo

Using a controlled experimental design, the study demonstrates the safety of a new brIFN-τ formulation administered intrauterine in Suffolk sheep. Microencapsulated hydrogel administration will be performed via intrauterine deposition using a standard transcervical artificial insemination device, without any invasive intervention. This route is expected to be routinely applicable in ruminants and will allow for the safe and efficient delivery of compounds directly into the uterus [93]. The formulation will be specifically designed to be compatible with this mode of application. The results indicate that formulation administration did not lead to significant behavioral, physiological, or hematological changes compared to the control group. The species assessments of rectal and vulvar temperatures, behavioral evaluations, and animal welfare analyses remained within normal ranges, showing no signs of pain, local inflammation, or acute adverse reactions. Red and white blood cell counts also confirmed the absence of treatment-induced infections or inflammatory processes. Previous studies have utilized formulations containing chitosan in hydrogel form in cows without observing adverse effects [90,94]. These findings collectively support the safety of the formulation and its delivery method, demonstrating that it does not compromise animal welfare or induce unwanted side effects. The result represents a crucial advancement for future clinical or research applications in ovine and bovine reproductive health.

A comprehensive analysis of the antiluteolytic activity of brIFN-τ formulation in non-pregnant cows was performed, utilizing three experimental groups with different compound formulations. Ultrasound observations revealed that the device with microencapsulated brIFN-τ in hydrogel (group 2) remained visible for a significantly extended period compared to the hydrogel with soluble brIFN-τ (group 3) (Figure 11B). This suggests a more prolonged and sustained release of brIFN-τ.

The findings correlate with progesterone levels, where group 2 maintained high and stable hormone concentrations throughout the estrous cycle. This result proved to have a more effective antiluteolytic effect than group 3 and the control group. A significant difference in progesterone levels observed on day 19 between group 2 and the control group indicates that microencapsulation enhances the ability of brIFN-τ to inhibit luteolysis and extend the estral cycle until day 25 at least (Figure 11D), compared to the 19-21 days without formulation [90]. In contrast, the non-encapsulated brIFN-τ shows limited effectiveness, likely due to its shorter release and action.

These results underscore the importance of formulation and delivery systems in enhancing the biological function brIFN-τ. This fact has significant implications for bovine reproductive management, particularly in preventing luteolysis and maintaining pregnancy.

## 5. Conclusions

In conclusion, the present study demonstrates that brIFN-τ can be efficiently produced in *Pichia pastoris*, purified, and structurally validated to ensure biological activity. The successful encapsulation of brIFN-τ in chitosan-based microparticles and its integration into starch–chitosan hydrogels has resulted in a novel sustained-release system suitable for intrauterine administration in cattle. In vitro assays confirmed that the encapsulated protein retained its antiviral activity, while in vivo studies showed that the formulation was safe and effective in modulating luteolysis. This fact was indicated by sustained progesterone levels and reduced corpus luteum regression. These findings suggest that microencapsulated brIFN-τ delivered through hydrogel matrices represents a promising biotechnological strategy to enhance maternal recognition of pregnancy, reducing early embryonic loss and improving reproductive performance in cattle. In future research, it would be valuable to compare the effectiveness of administering exogenous progesterone in cows with the use of an intrauterine device containing interferon tau, particularly regarding pregnancy success rates in cattle. This comparison could be especially relevant in systems that utilize estrus synchronization and artificial insemination. Additionally, we could evaluate the impact of our microencapsulated formulation when administered via injection in animals and investigate its effects on pregnancy outcomes. These suggestions for future investigations are based on the results obtained in this study.

## 6. Patents

The patent granted in Chile No. C-2021-01591 resulted from the work reported in this work. Registration No.: 68017, INAPI, Chile.

## Figures and Tables

**Figure 1 biomolecules-15-01009-f001:**
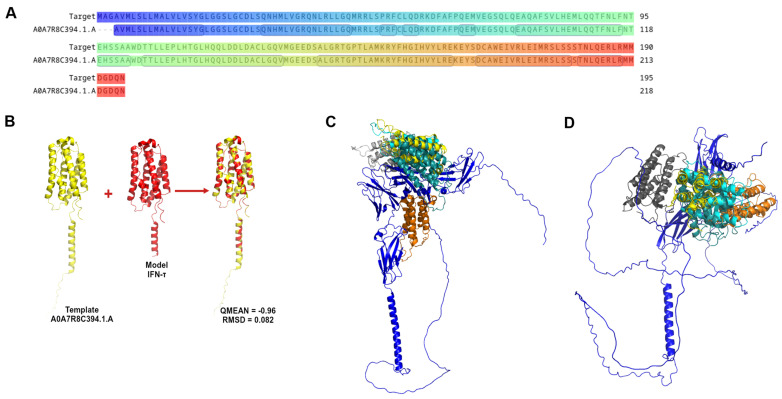
Graphic representation of IFN-τ sequence and homology modelling. (**A**) Alignment of the coiled IFN-τ (target: G3MZ51·G3MZ51_BOVIN from UniProt) and the Template A0A7R8C394.1.A. (**B**) The best model obtained from the Homology Modelling and Docking Experiment. (**C**) Represents the best poses of α/β-Receptor-1 (blue) with interferon tau. (**D**) Represents the best poses of α/β-Receptor-2 (blue) with IFN-τ.

**Figure 2 biomolecules-15-01009-f002:**
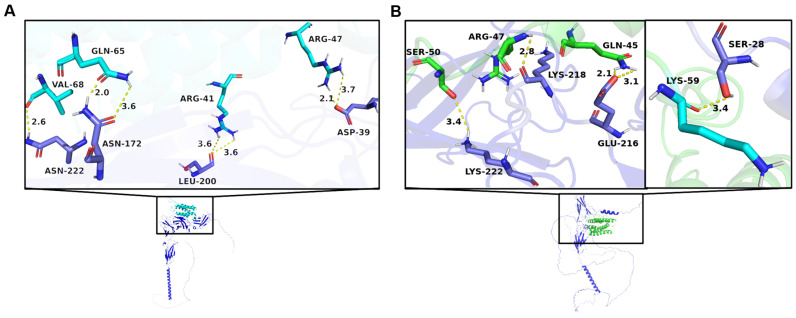
Interaction between IFN-τ and receptors. (**A**) Graphical representation of the main interactions between IFN-τ and α/β-Receptor-1. (**B**) Graphical representation of the main interactions between IFN-τ and α/β-Receptor-2. The amino acids of α/β-Receptor-1 and 2 are shown in dark blue.

**Figure 3 biomolecules-15-01009-f003:**
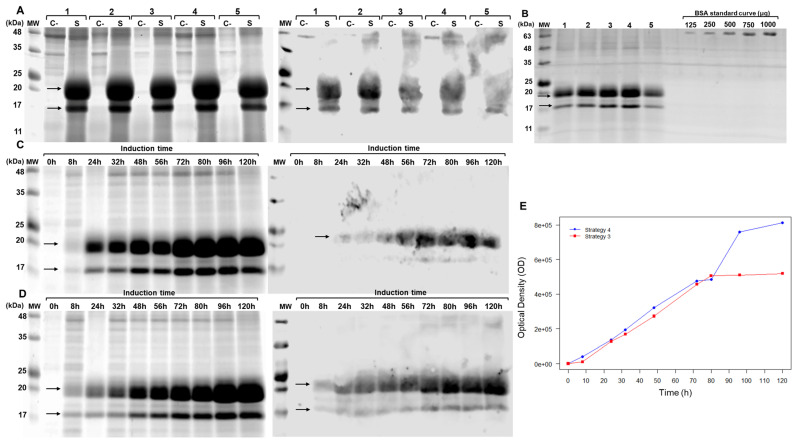
Evaluation of brIFN-τ expression in *Pichia pastoris*. (**A**) SDS-PAGE (right) and Western blot (left) of the *Pichia pastoris* culture supernatant collected 72 h post-induction. Samples are labeled as follows: 1: strategy 1, 2: strategy 2, 3: strategy 3, 4: strategy 4, 5: strategy 5, C-: negative control, S: sample. The arrows indicate the bands corresponding to recombinant bovine IFN-τ. (**B**) SDS-PAGE gel was used to quantify brIFN-τ from the Pichia pastoris culture supernatant, which was collected 72 h post-induction. Lanes 1–5 represent culture supernatants from strategies 1–5. A BSA standard curve (μg) was used: 125, 250, 500, 750, and 1000 μg. MW: molecular weight markers (kDa). (**C**) Increasing the culture incubation time to 120 h in shaken flasks. This section includes an SDS-PAGE (right) and Western blot (left) of the culture supernatant using induction strategy 3 at various time points (0–120 h). (**D**) SDS-PAGE (right) and Western blot (left) using strategy 4 at various time points (0–120 h). (**E**) brIFN-τ expression curve plotted as a function of time and optical density (OD) for strategies 3 and 4. Original figures can be found in Appendix A.

**Figure 4 biomolecules-15-01009-f004:**
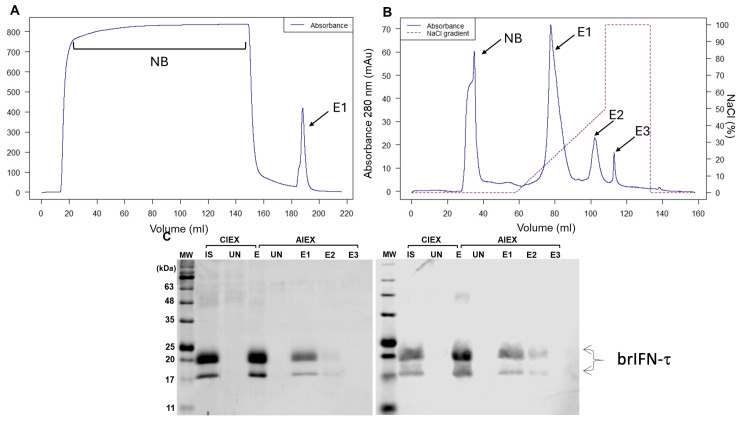
Evaluation of brIFN-τ purification. (**A**) Chromatogram from cationic exchange. (**B**) Chromatogram from anionic exchange. MW: molecular weight, CIEX: cation exchange chromatography, AIEX: anion exchange chromatography, IS: initial sample, UN: unbound proteins to the column, E1: first elution, E2: second elution, E3: third elution fraction. The NaCl gradient used to elute the protein fractions is represented in the graph by a dashed line. (**C**) SDS-PAGE gel (right) and Western blot (left) of fractions obtained from cationic exchange (S) and anionic exchange (Q) purification. Original figures can be found in Appendix A.

**Figure 5 biomolecules-15-01009-f005:**
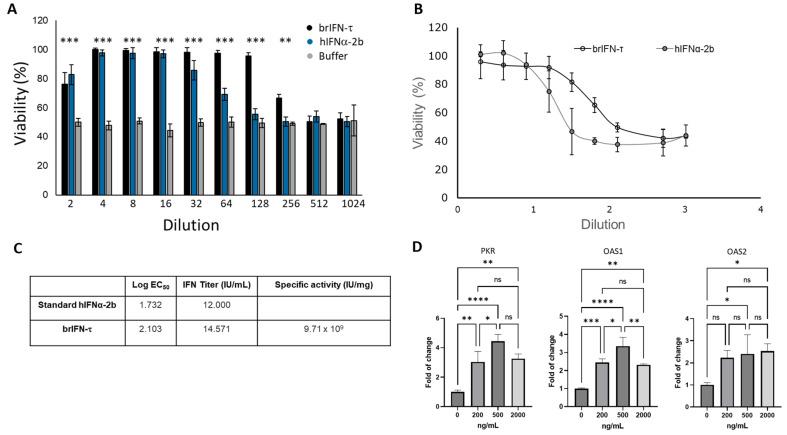
Evaluation of the antiviral activity of brIFN-τ. (**A**) Percentage of cell viability in inhibiting the cytopathic effect of the Mengo virus in MDBK cells. (**B**) Antiviral activity of brIFN-τ, with IFNα-2b used as a standard for comparison. (**C**) Titers of brIFN-τ and specific activity of the standard. (**D**) As determined by qPCR, relative expression levels of PKR, OAS1, and OAS2. Data were normalized using the Pfaffl method [59,60] with β-actin as the housekeeping gene. Data were statistically analyzed using one-way analysis of variance and Dunnett’s post-test (**** *p* < 0.0001 *** *p* < 0.001, ** *p* < 0.01, * *p* < 0.05) (ns: *p* > 0.05).

**Figure 6 biomolecules-15-01009-f006:**
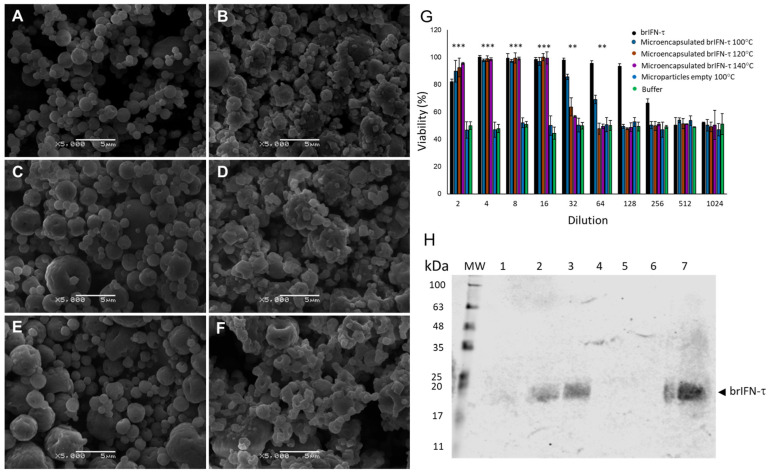
Microparticle characterization, viability and brIFN-t release. Scanning electron microscope (SEM) photomicrographs of microparticles generated at different temperatures are presented (**A**–**F**). The empty control particles, which do not contain brIFN-t, were produced at temperatures of 100 °C (**A**), 120 °C (**C**), and 140 °C (**E**). In contrast, brIFN-t associated chitosan particles were generated at the same temperatures: 100 °C (**B**), 120 °C (**D**), and 140 °C (**F**). The viability percentage was assessed by measuring the inhibition of the cytopathic effect of the Mengo virus in MDBK cells (**G**). A Western blot analysis of microparticle release samples was conducted at 100 °C, 120 °C, and 140 °C (**H**). The PPM serves as a molecular weight standard, and lanes 1 to 3 correspond to the release supernatants of particle samples generated at 100 °C, 120 °C, and 140 °C, respectively, after 48 h of release in a 10 mM citrate solution at pH 6.5 and 37 °C. Lanes 4 to 6 represent the release samples of empty microparticles generated at 100 °C, 120 °C, and 140 °C, respectively. Lane 7 shows the brIFN-t used as a control (20 µg). A polyclonal antibody from Santa Cruz (Fermelo S.A, Santiago de Chile, Providencia, Chile) was utilized in the experiment, and the reaction was visualized using the Alexa 680 rabbit anti-IgG secondary antibody (Thermo Fisher Scientific, Waltham, MA, USA) Data were statistically analyzed using one-way analysis of variance and Dunnett’s post-test (*** *p* < 0.001 and ** *p* < 0.01). Original figures can be found in Appendix A.

**Figure 7 biomolecules-15-01009-f007:**
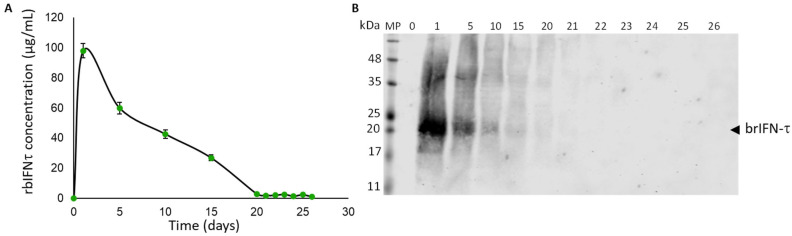
Graphical representation of the daily release of brIFN-τ from microparticles generated at 120 °C. (**A**) Total protein quantification from the released samples daily in the resuspension buffer (50 mM citrate buffer, pH 6.8 and 38 °C to simulate the uterine environment) was performed using the micro-BCA™ Protein Assay Kit. (**B**) A Western blot analysis was conducted to determine the release of brIFN-τ from chitosan microparticles. The lanes represent the days supernatant samples containing brIFN-τ were extracted. A polyclonal anti-IFN-τ antibody produced in rabbits (Santa Cruz) was utilized, and the reaction was visualized using a secondary antibody, anti-rabbit IgG Alexa 680. The Li-COR Biosciences Odyssey scanner system, along with Image Studio Lite version 5.2 analysis software, was used for visualization and analysis. Original figures can be found in Appendix A.

**Figure 8 biomolecules-15-01009-f008:**
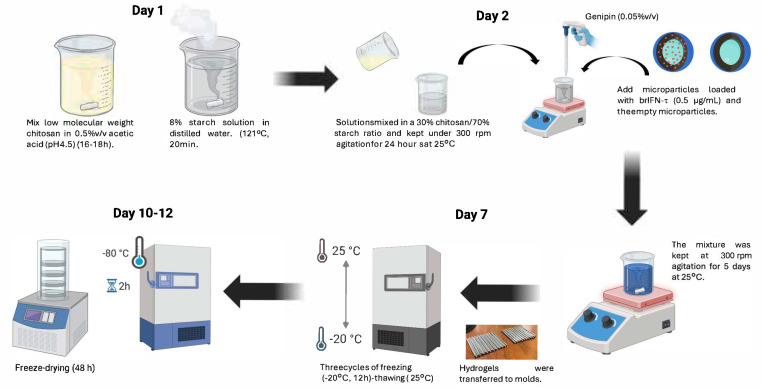
Scheme of hydrogel generation.

**Figure 9 biomolecules-15-01009-f009:**
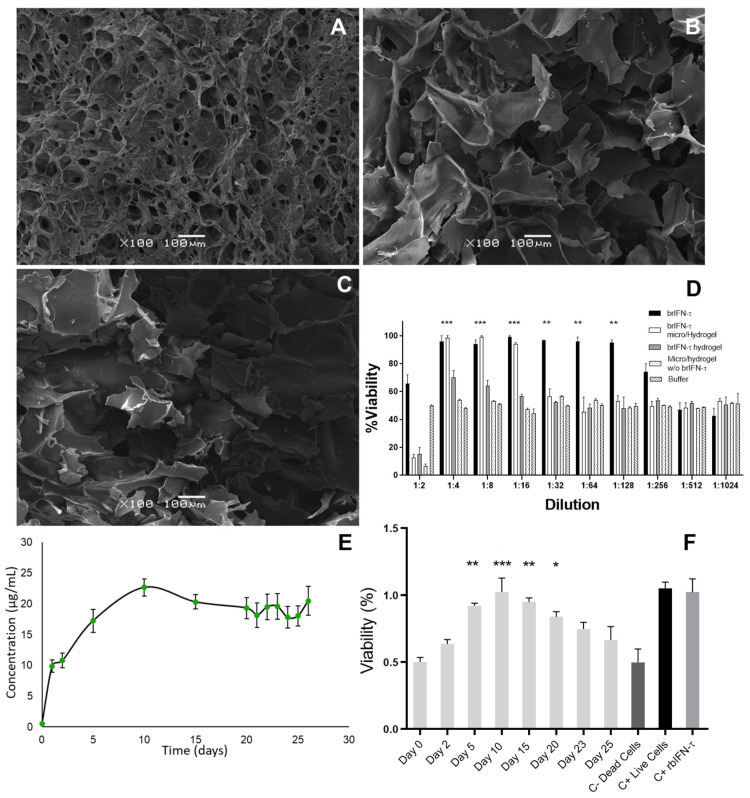
Hydrogel characterization and brIFN-τ release assay. Photomicrographs were taken using scanning electron microscopy (SEM) of the hydrogel samples. (**A**) Hydrogel mixed with an equivalent of 2 mg of brIFN-τ microencapsulated. (**B**) Hydrogel mixed with 2 mg of non-microencapsulated soluble brIFN-τ. (**C**) Hydrogel mixed with empty microparticles. All microparticles were prepared at 120 °C. The brIFN-τ, both microencapsulated and soluble, was added on day 2 of the hydrogel elution scheme, immediately after adding genipin. (**D**) Viability percentage was determined by measuring the inhibition of the cytopathic effect of the Mengo virus assay in MDBK cells using samples released 48 h after each sample preparation. (**E**) A graphical representation illustrates the daily release of hydrogel mixed with brIFN-τ microencapsulated over a period of 26 days. Total protein quantification was achieved using the Micro BCA™ Protein Assay Kit. The x-axis corresponds to the day of sampling. (**F**) This section represents the antiviral effect of brIFN-τ in MDBK cell cultures using ½ dilutions. The samples of hydrogel containing brIFN-τ microencapsulated were utilized for this assay. Results were compared to a negative control of dead cells. Data were statistically analyzed using one-way analysis of variance and Dunnett’s post-test (*** *p* < 0.001, ** *p* < 0.01, * *p* < 0.05).

**Figure 10 biomolecules-15-01009-f010:**
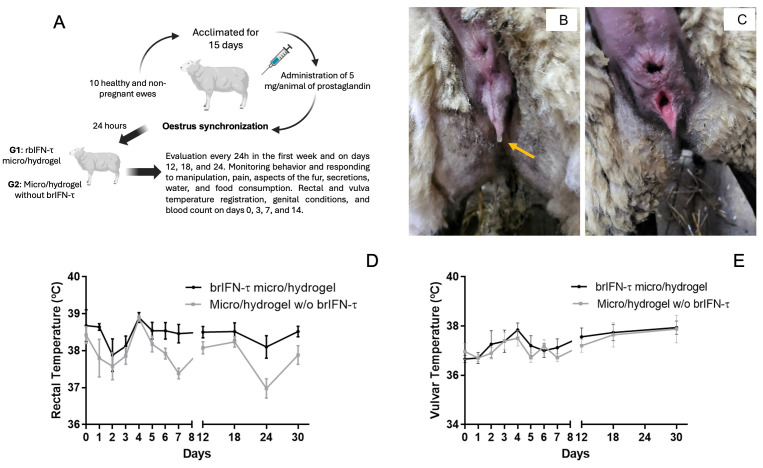
Drug safety evaluation in the ovine model. (**A**) Schema of the ovine assay. (**B**) External evaluation of the genitalia in the study ewes at time 0. Edema and mucosal secretion (yellow arrow). (**C**) Vulvar edema and erythematous mucosa are due to estrus. (**D**) Evolution of the rectal temperature in ewes treated with brIFN-τ microencapsulated in hydrogel (brIFN-τ micro/hydrogel) v/s the control group (micro/hydrogel w/o brIFN-τ). (**E**) Variation in the vulvar temperature in ewes was treated in the same way. Results were compared concerning the negative control of dead cells. No significant differences were registered between the groups.

**Figure 11 biomolecules-15-01009-f011:**
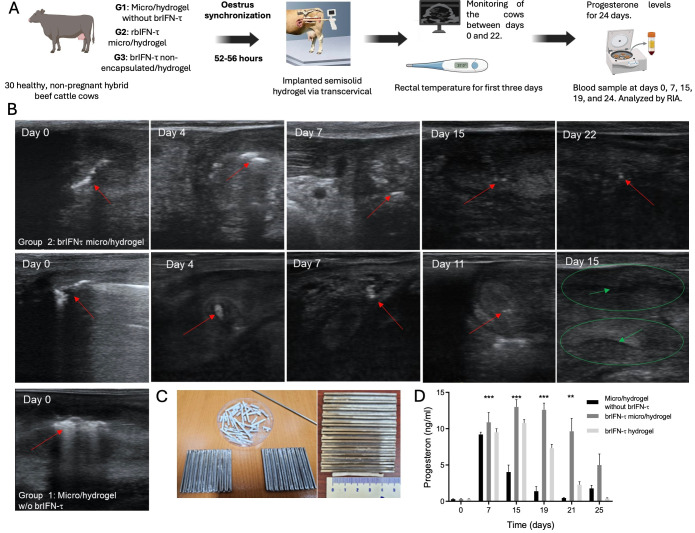
Anti-luteolytic activity and serum progesterone levels after brIFN-τ delivery via hydrogel in cattle. (**A**) Scheme of progesterone assay in cows. (**B**) Transrectal ultrasound evaluation of cows from groups 1, which used microparticles/hydrogel without brIFN-τ, 2 (brIFN-τ microencapsulated with hydrogel), and 3 (non-encapsulated brIFN-τ plus, hydrogel). Visible implant in uterine horn (red arrow); uterine horn (green circle); anechoic center of the uterine horn, indicating the absence of solid contents (green arrow). Ultrasonograph Mindray^®^, DP50-Vet, linear probe 8.5 MHz. (**C**) Mold used to generate semi-solid matrix (hydrogel) and hydrogel examples. Sample of implant length (4 cm) (0.3 cm diameter). (**D**) Progesterone (P4) levels in the three groups: control, brIFN-τ micro/hydrogel, and brIFN-τ hydrogel (nonencapsulated). The levels of P4 on day 19 stand out, with significant differences between brIFN-τ micro/hydrogel concerning the control group. Data were statistically analyzed using one-way analysis of variance and Dunnett’s post-test (*** *p* < 0.001 and ** *p* < 0.01).

**Table 1 biomolecules-15-01009-t001:** Methanol concentration per induction strategy over time.

Strategy	0 h	24 h	48 h
1	0.3%	0.4%	0.5%
2	0.5%	0.75%	1%
3	0.5%	1%	1.5%
4	0.5%	1%	1%
5	1%	1%	1%

**Table 2 biomolecules-15-01009-t002:** Optimal strategies for methanol concentration in batch ulture.

Strategy	0 h	24 h	48 h	72 h	96 h
3	0.5%	1%	1.5%	1.5%	1.5%
4	0.5%	1%	1%	1%	1%

**Table 3 biomolecules-15-01009-t003:** Description of the primers used.

Gene Name-Gene ID (NCBI Gene)	Forward Primer (5′-3′)	Reverse Primer (5′-3′)	Tm	Product Size (bp)
Bos taurus 2′-5′-oligoadenylate synthetase 1 (OAS1) mRNA-NM_001040606.1	AAATAGCTGGGAGCGGCTTG	CTGTGTTCTTGGGGCGACAC	60	114
Bos taurus 2′-5′-oligoadenylate synthetase 2 (OAS2), mRNA-NM_001024557.1	GCCTTCAATGCTCTGGGCTT	CAGGCCTGGCTTTCACCATA	58	90
Bos taurus eukaryotic translation initiation factor 2 alpha kinase 2 (PKR), mRNA-NM_178109.3	TGGAGACACGGAAGAGCTGT	ATGTCACGGACGATTTCCGC	56	122
Bos taurus beta-actin (ACTB) mRNA-AY141970.1	GCCCATCTATGAGGGGTACG	ATGTCACGGACGATTTCCGC	60	147

**Table 4 biomolecules-15-01009-t004:** Microencapsulation of chitosan control and chitosan/brIFN-τ mixture. Spray drying of vacuum chitosan samples in the mini-spray dryer b-290 at different temperatures, with constant feed, aspiration, and airflow.

Inlet Temperature (°C)	Inlet Temperature (°C)	Aspiration (%)	Feed Flow Rate (mL/min)	Air Flow Rate (L/h)
100	63	95	6.0	536
120	76	95	6.0	536
140	88	95	6.0	536

**Table 5 biomolecules-15-01009-t005:** Template search results for building the IFN-τ model.

Template	Seq Identity (%)	Found by	Method	Seq Similarity	Coverage	Description
A0A7R8C394.1.A	100.00	AFDB	AlphaFold v2	0.61	0.98	Interferon 1BE10
1b5l.1.A	68.82	BLAST	X-ray	0.50	0.87	Interferon Tau
1b5l.1.A	68.64	HHblits	X-ray	0.50	0.87	Interferon tau
3oq3.1.A	53.61	HHblits	X-ray	0.45	0.85	Interferon alpha-5
3se4.1.B	66.08	HHblits	X-ray	0.50	0.88	Interferon omega-1
3se4.1.B	66.07	BLAST	X-ray	0.50	0.86	Interferon omega-1
3oq3.1.A	54.94	BLAST	X-ray	0.45	0.83	Interferon alpha-5
2kz1.1.A	56.10	HHblits	NMR	0.46	0.84	Interferon alpha-2
7e0e.1.A	50.30	HHblits	X-ray	0.43	0.86	Interferon alpha-2
7e0e.1.A	51.85	BLAST	X-ray	0.44	0.83	Interferon alpha-2
2lag.1.B	56.10	HHblits	NMR	0.46	0.84	Interferon alpha-2
4z5r.1.A	56.10	HHblits	X-ray	0.46	0.84	Interferon alpha-2
1itf.1.A	56.10	HHblits	NMR	0.46	0.84	Interferon Alpha-2a
2hym.1.B	56.10	HHblits	NMR	0.46	0.84	Interferon alpha-2
3ux9.1.A	57.67	BLAST	X-ray	0.46	0.84	Interferon alpha-1/13
1au1.1.A	34.15	HHblits	X-ray	0.36	0.84	Interferon-Beta
3ux9.1.A	56.63	HHblits	X-ray	0.45	0.85	Interferon alpha-1/13
3se3.1.B	54.55	HHblits	X-ray	0.45	0.85	Interferon alpha 2b
3s9d.1.A	54.55	HHblits	X-ray	0.45	0.85	Interferon alpha-2
1au1.1.B	34.15	HHblits	X-ray	0.36	0.84	Interferon-Beta
2lms.1.A	55.76	HHblits	NMR	0.45	0.85	Interferon alpha-2
6jhd.1.A	56.36	HHblits	NMR	0.45	0.85	Interferon alpha-8
6jhd.1.A	57.06	BLAST	NMR	0.45	0.84	Interferon alpha-8
1au1.1.A	37.50	BLAST	X-ray	0.39	0.70	Interferon-beta
1au1.1.B	37.50	BLAST	X-ray	0.39	0.70	Interferon-beta

**Table 6 biomolecules-15-01009-t006:** Docking and confidence scores of top five IFN-τ–α/β receptor complex poses predicted by HDOCK.

Poses	IFN-τ-α/β-Receptor-1		IFN-τ-α/β-Receptor-2	
	Docking Score	Confidence Score	Docking Score	Confidence Score
Pose-1	−259.84	0.9000	−271.82	0.9196
Pose-2	−249.51	0.8798	−270.79	0.9180
Pose-3	−240.63	0.8597	−266.38	0.9111
Pose-4	−238.00	0.8532	−256.82	0.8944
Pose-5	−236.13	0.8485	−251.72	0.8844

**Table 7 biomolecules-15-01009-t007:** Chitosan/brIFN-τ encapsulation performance at varying temperatures under constant feed, suction, and airflow.

Sample	Mass (g)	Yield (%)
100 °C	0.96	56.1
120 °C	1.14	66.1
140 °C	1.16	67.8

**Table 8 biomolecules-15-01009-t008:** Performance of empty chitosan encapsulation at varying temperatures with constant feed, suction, and airflow.

Sample	Mass (g)	Yield (%)
100 °C	0.96	56.1
120 °C	1.14	66.1
140 °C	1.16	67.8

**Table 9 biomolecules-15-01009-t009:** Hemogram results in sheep on days 0, 3, 7, and 14 of the safety trial.

	Hematocrit				Total Leukocytes			
Group	Day 0	Day 3	Day 7	Day 14	Day 0	Day 3	Day 7	Day 14
Control Group	10.16 ± 0.46 a	10.48 ± 0.83 a	10.47 ± 0.95 a	10.66 ± 0.95 a	6876 ± 1443 a	8560 ± 2211 a	10,148 ± 4071 a	8926 ± 2749 a
Treated Group	9.37 ± 1.08 a	10.02 ± 1.35 a	10.23 ± 1.36 a	9.84 ± 1.18 a	6374 ± 2834 a	7440 ± 2721 a	8306 ± 2513 a	7082 ± 2906 a

The means and standard deviations (SD) relating to the two groups of five animals per group are represented. Same letters in the cells of the columns (a) indicate non-significant differences (*p* > 0.05) in the indicator between the treated group and the control group.

## Data Availability

The original contributions presented in this study are included in the article. Further inquiries can be directed to the corresponding author.

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
