# Peer review of "A Novel Microencapsulated Bovine Recombinant Interferon Tau Formulation for Luteolysis Modulation in Cattle"

_biomolecules, 2025, doi:10.3390/biom15071009_

Round 1
Reviewer 1 Report
Comments and Suggestions for Authors
The article titled "A novel microencapsulated bovine recombinant interferon Tau formulation for luteolysis modulation in cattle" was reviewed. The topic of the article is novel and the article is well written, but there are some points that should be clarified in the article. These tips are provided below.
Abstract
Summarize the descriptions related to the introduction section and add more detailed information related to the materials and methods section as well as the results section.
Introduction
Introduction should include more details about the maternal recognition of pregnancy in cattle, especially the molecular aspects. Delete some unnecessary information about the PGF2α and add some more details about the molecular aspects of IFN-τ roles in the maternal recognition of pregnancy in cattle.
Materials and Methods:
2.14. Drug safety in the ovine model
Describe the estrus synchronization protocol exactly and what was the purpose of synchronization.
Considering the presence of cervical rings in ewes, describe the exact protocol that was used for passing the straws completely through the cervix.
Because the drug has injected into the uterus, assessment of drug safety should be focused on the uterus. Pathological changes in the uterus usually do not show general systemic symptoms. Describe about it.
2.15. Anti-luteolytic activity in cows 375
“Ultrasonographic monitoring in the cows was performed seven times between days 0 and 22 to determine the degradation rate of the implant”. Describe about the method of evaluation to determine the degradation rate of the implant.
“Besides, during the first three days after the implant's administration, each animal's rectal temperature was recorded to assess the 392 safety of the treatments”. Similar to the previous comment, assessment of drug safety should be focused on the uterus. Pathological changes in the uterus usually do not show general systemic symptoms. Describe about it.
Discussion
The question that arises is that given that the purpose of producing microencapsulated bovine recombinant IFN-τ, is using in inseminated cows to increase the chance of pregnancy, and this compound was used in this study as an intrauterine injection, a method that makes its use in inseminated cows impractical, it is necessary for the authors to explain this in the article and the method of practical applying of this compound in the future.
Comments on the Quality of English Language
The quality of the English writing is generally good, but could be improved with some editing.
Author Response
|
3. Point-by-point response to Comments and Suggestions for Authors |
|
Comments 1: Abstract Summarize the descriptions related to the introduction section and add more detailed information related to the materials and methods section as well as the results section. |
|
Response 1: Thank you for your valuable suggestion. We agree with this comment. Accordingly, we have revised the Abstract to provide a more concise introduction and to include more specific details regarding the materials and methods as well as the main findings of the study. The updated abstract now summarizes the experimental design, including the expression system, encapsulation strategy, biological assays, and key in vivo results. These changes can be found in the revised manuscript on page 1, paragraph 1. |
|
Comments 2: Introduction Introduction should include more details about the maternal recognition of pregnancy in cattle, especially the molecular aspects. Delete some unnecessary information about the PGF2α and add some more details about the molecular aspects of IFN-τ roles in the maternal recognition of pregnancy in cattle. |
|
Response 2: Thank you for pointing this out. Agree. We have, accordingly modified the introduction to emphasize this point. We deleted the unnecessary information about PGF2alfa and, instead, focused on including more details about maternal recognition of pregnancy with focus on the molecular aspects, as well as adding more references to explain it. The updated text can be found on page 2, paragraph 2, lines 63-89.
|
|
Comments 3: 2.14. Drug safety in the ovine model Describe the estrus synchronization protocol exactly and what was the purpose of synchronization. |
|
Response 3: Thank you for pointing this out. Agree. We have, accordingly, modified the text and added the protocol to emphasize this point. The protocol begins with administering 5 mg of intramuscular prostaglandin to the ewe at 0. After seven days, a second dose of prostaglandin is given. Twenty-four hours later, when estrus is detected, the implant is applied. This procedure ensures that estrus occurs when administering the interferon implant, which would typically coincide with mating or insemination. By doing this, the cervix remains open, allowing for easier insertion of the injector to place the implant in the uterus. Said description can be found on page 9, paragraph 1 and lines 354-360.
Comments 4: 2.14. Drug safety in the ovine model Considering the presence of cervical rings in ewes, describe the exact protocol that was used for passing the straws completely through the cervix. Response 4: Thank you for pointing this out. Agree. We have, accordingly, explained this protocol to emphasize this point. The procedure was conducted during the estrus period when the cervix was open and dilated, as indicated in the figure with speculum (d). We considered including this information as supplementary material but ultimately decided against it to avoid overwhelming the article with additional details. Additionally, we used epidural anesthesia, as described in the text, which made it unnecessary to force the passage of the injector.
Sequence of the procedure of administration of the intrauterine interferon. (a) Epidural anesthesia. (b) Charge of the dispositive in the injector. (c) Ewe restrained in the gynecological table. (d) Observation of the genitalia through the speculum, with open cervix and normal mucosal secretion of an estrus condition. (e) Insertion of the injector in the cervix through the speculum. (f) Injection of the intrauterine formula. (g) Removal of the injector. If you consider it we can add it as supplementary material.
Comments 5: 2.14. Drug safety in the ovine model Because the drug has injected into the uterus, assessment of drug safety should be focused on the uterus. Pathological changes in the uterus usually do not show general systemic symptoms. Describe about it. Response 5: Thank you for pointing this out. Agree. We have, accordingly, explained this to emphasize this point. This study aimed to evaluate the acute reactivity to the administered product (which was not injected), as it could potentially trigger a systemic inflammatory response. Monitoring this parameter for new formulations, even those applied locally, is essential. Local reactions were assessed by observing signs of inflammation, such as redness, swelling, pain, increased temperature, and any secretions; however, no such reactions were noted in any of the animals. This was particularly relevant as the animals were in estrus for up to three days, during which we would expect to see any secretions if they were present. For a precise analysis of the effects on the uterus, alternatives such as hysterectomy or euthanasia for histopathological evaluation of the tissues were considered, but these methods were ultimately ruled out for ethical reasons. |
|
Comments 6: 2.15. Anti-luteolytic activity in cows 375 “Ultrasonographic monitoring in the cows was performed seven times between days 0 and 22 to determine the degradation rate of the implant”. Describe about the method of evaluation to determine the degradation rate of the implant. Response 6: Thank you for pointing this out. Agree. We have, accordingly, modified the term degradation for disintegration to emphasize this point. We do not refer to this as a degradation rate. Instead, we have changed the term to "degradation by disintegration" in the document. What we actually did was visually monitor the implant in the uterus to determine how long it remained visible by ultrasound in the animal. This device is highly hydrophilic, meaning it initially absorbs liquid and gradually disintegrates, allowing for a prolonged release of interferon. This change can be found on page 10, paragraph 3, and line 396.
Comments 7: 2.15. Anti-luteolytic activity in cows 375 “Besides, during the first three days after the implant administration, each animal's rectal temperature was recorded to assess the 392 safety of the treatments”. Similar to the previous comment, assessment of drug safety should be focused on the uterus. Pathological changes in the uterus usually do not show general systemic symptoms. Describe about it. Response 7: Thank you for pointing this out. Agree. We have, accordingly, explained to emphasize this point. In a manner analogous to the assessments conducted in drug safety studies involving ovine models, local reactions were evaluated by monitoring signs of inflammation, including redness, swelling, pain, elevated temperature, and any secretions. However, no such reactions were observed in any of the animals. In this study involving cows, ultrasound evaluations indicated no alterations in the uterus, such as inflammatory responses or the presence of retained secretions.
Comments 8: Discussion The question that arises is that given that the purpose of producing microencapsulated bovine recombinant IFN-τ, is using in inseminated cows to increase the chance of pregnancy, and this compound was used in this study as an intrauterine injection, a method that makes its use in inseminated cows impractical, it is necessary for the authors to explain this in the article and the method of practical applying of this compound in the future. Response 8: Thank you for your observation. We agree that it is important to clarify the route of administration used. The microencapsulated compound was delivered intrauterinely using the same transcervical device commonly employed in bovine artificial insemination (standard 4 mm straw), and not through an invasive intrauterine injection. This method is practical, accessible, and routinely applied in reproductive protocols. To avoid misunderstandings, we have incorporated this clarification into the Discussion section (page 28, paragraph 1, lines 960–965), also noting that this formulation was designed to be compatible with standard field insemination procedures.
The quality of the English writing is generally good, but could be improved with some editing. 4. Response to Comments on the Quality of English Language |
|
Point 1: |
|
Response 1: Thank you for your observation. We have taken it into careful consideration. The manuscript has been revised for clarity, grammar, and academic style, with improvements implemented throughout the Abstract, Introduction, Results, and Discussion sections. Particular attention was paid to sentence structure, word choice, and consistency in terminology to enhance the overall readability and scientific precision of the text. |
|
5. Additional clarifications |

Reviewer 2 Report
Comments and Suggestions for Authors
PLEASE IGNORE THE ATTACHED FILE AS I WOULD LIKE TO REPLACE IT WITH THE CURRENT VERSION BUT THE SYSTEM WON'T REMOVE THE PREVIOUS VERSION.
Comments to authors
This manuscript presents a multidisciplinary study developing a novel sustained-release intrauterine formulation of bovine recombinant interferon tau (brIFN-τ) to improve luteal maintenance and reduce early embryonic loss in cattle. The integration of bovine recombinant interferon tau with chitosan-based nanoparticles and in vivo evaluation in ovine and bovine models is ambitious and well-aligned with the study goals. This study is detailed and may need to be shortened for the reader to improve clarity, scientific rigor, and interpretive depth. The revision details are noted below.
- Abstract: Please provide comparable between this formulation and the existing method, also it would be nice to include brief mention of the unique combination of chitosan encapsulation and hydrogel delivery.
- Introduction: It would be nice to explain why the alternative strategy is needed in term of embryo signaling and maternal recognition.
- Methods:
- There are overly detailed in this section. Some sections which is not related to the biological implications are better to move to the supplementary section.
- Ultrasound alone may not sufficient to describe the hydrogel degradation. Please clarify whether another methods were assessed in vivo.
- Results:
- Please explain why use Mengo virus instead of other virus (eg. BVDV that impact on the reproduction) in this study as this virus is uncommon in reproductive research.
- In Table 8, it would be nice to report group means ± SD rather than raw individual valuesand specifically mentioned any statistically or clinically relevant changes.
- Figure 11A: please fix the non-english wording such as “52-56 horas” and “19y24”.
- Figure 11B: please fix the typhos “group 1: Micro/hidrogel”.

Author Response
|
3. Point-by-point response to Comments and Suggestions for Authors |
|
Comments 1: Abstract Please provide comparable between this formulation and the existing method, also it would be nice to include brief mention of the unique combination of chitosan encapsulation and hydrogel delivery. |
|
Response 1: Thank you for pointing this out. We agree with this comment. Therefore, we have added your observation to the Abstract. The changes can be found on page 1 in paragraph 1:
|
|
Comments 2: Introduction It would be nice to explain why the alternative strategy is needed in term of embryo signaling and maternal recognition. |
|
Response 2: Thank you for pointing this out. Agree. We have, accordingly, modified the introduction to emphasize this point. The modifications suggested in the introduction were made. The changes can be found on page 2 on paragraph 2 lines 66-68.
|
|
Comments 3: Methods There are overly detailed in this section. Some sections which is not related to the biological implications are better to move to the supplementary section. |
|
Response 3: Thank you for pointing this out. Agree. We have, accordingly, discussed this point with the editor to emphasize it.
Comments 4: Methods Ultrasound alone may not sufficient to describe the hydrogel degradation. Please clarify whether another methods were assessed in vivo. Response 4: Thank you for pointing this out. Agree. We have, accordingly, explained to emphasize this point. The implant appeared as a hyperechoic structure on the initial ultrasound (time 0), prompting follow-up assessments to determine its visibility duration. The alternative approach would have involved serial euthanasia to locate the implant within the uterus, which was neither practical nor ethically justifiable. Consequently, progesterone levels were monitored.
Comments 5: Result Please explain why use Mengo virus instead of other virus (eg. BVDV that impact on the reproduction) in this study as this virus is uncommon in reproductive research. Response 5: Thank you for pointing this out. Agree. I/We have, accordingly, revised to emphasize this point. We used the Mengo virus instead of a virus that impacts reproduction (BVDV) because the objective of this assay is not to determine reproductive effect. Since interferon-tau is a type 1 interferon, we can use a cytopathic effect reduction assay on bovine cells, in this case, MDBK, a culture we have in the laboratory, to determine if the IFN is active. In this way, we can evaluate the antiviral activity of interferon-tau and follow its correct folding in the different steps of obtaining the final product. The Mengo virus is a cytopathic virus widely used for these assays. However, assays using BVDV, a cytopathic strain, could be designed to determine antiviral activity. |
|
Comments 6: Result In Table 8, it would be nice to report group means ± SD rather than raw individual values, and specifically mention any statistically or clinically relevant changes. Response 6: Thank you for pointing this out. Agree. We have, accordingly, changed the table to emphasize this point. In Table 9, the means ± SD are represented, and letters were used to indicate the significant differences between the treated group and the control group for hematocrit and total leukocytes. This change can be found on page 22, Table 9, paragraph 3, and lines 728-730. Page 23 paragraph 2 Lines 753 to 760 explain these results. Comments 7: Result Figure 11A: please fix the non-english wording such as “52-56 horas” and “19y24”. Response 7: Thank you for pointing this out. Agree. We have, accordingly, changed to emphasize this point. This change can be found in Figure 11A.
Comments 8: Result Figure 11B: please fix the typhos “group 1: Micro/hidrogel”. Response 8: Thank you for pointing this out. Agree. I/We have, accordingly, changed to emphasize this point. This change can be found in Figure 11B.
The quality of the English writing is fine and does not requeri ay improvement 4. Response to Comments on the Quality of English Language |
|
Point 1: |
|
Response 1: Thanks for your observation |
|
5. Additional clarifications |
|
|

Reviewer 3 Report
Comments and Suggestions for Authors
See file added

Verbal timing to be revised
Author Response
|
3. Point-by-point response to Comments and Suggestions for Authors Comments: |
|
Very well done study with a remarkable experimental protocol. The only real problem is the verbosity of the Results section, in this section the references already presented in Materials and Methods or to be presented in Conclusions should not be re-presented because all this makes it really annoying to read. In consideration of the fact that I consider the publication of this study in Biomolecules important, because it would increase the scientific validity of the journal, I subordinate the acceptance of the study to the rewriting of the Resuts section, as reported below. I remind the authors that what is not statistically validated should not be considered. Major revision: 3.1.1. Homology Modeling The starting point for homology modeling is reported in Fig. 1A. Based on this sequence, a comprehensive search for suitable structural templates was 430 conducted to construct homology-based 3D models. A total of 25 candidate templates 431 were evaluated (Table 4). The structure with a QMEAN value closest to 0 and the lowest RMSD 453 was selected among the generated models. These criteria indicated high structural accuracy and consistency with the reference template. The chosen model accurately represents the three-dimensional architecture of bovine IFN-τ (Fig. 1B). The model displayed structural features consistent with functional interferon, supporting its suitability for further structural and interaction analyses. 3.1.2. Docking assays The top five binding poses with their corresponding Docking Scores and Confidence Scores for both receptor complexes are reported in Table 5. The conformations had a Docking Score below -200 kcal/mol, indicating that the complexes between bovine IFN-τ and α/β-Receptors-1 and 2 are stable (Table 5). However, the most negative docking score was found in the complexes formed with IFN-τ and α/β- Receptor-2, indicating a higher affinity between these two proteins. The docking experiments show that the interaction between IFN-τ and its type-1 receptor generated seven hydrogen bond interactions, only three smaller than 3 Å (Fig. 2). These interactions are also strong and capable of stabilizing a protein-ligand complex. In this complex (IFN-τ-α/β-Receptor-1), the strongest interactions were found between the carbonyl oxygen of Gln65 of IFN-τ with the NH2 of Asn172 of the receptor at 2.0 Å and the NH2 of the side chain of Arg47 of IFN-τ with the carboxyl group of the side chain of Asp39 of the receptor at 2.1 Å. Another strong interaction was found between the carboxyl group of Val68 of IFN-τ and the NH2 group of Asn222 of the receptor at 2.6 Å (Fig. 2A). The strongest interactions were found between the amino group of Gln45 of IFN-τ with the carboxyl-terminal of Glu216 of the receptor at 2.1 Å and the NH2 group of the functional group of Arg47 of IFN-τ with the carboxyl group of Lys218 of the receptor at 2.8 Å (Fig. 2B). The other interaction considered strong in this complex, is the other nitrogen of the amino group of Gln45 of interferon with the carboxyl-terminal of Glu216 at 3.1 Å. These two hydrogen bonds formed by Gln45 of interferon confer a high stability to this complex 3.2.1. brIFN-τ expression SDS-PAGE analysis of culture supernatants revealed two protein bands of approximately 17 kDa and 20 kDa, absent in the negative control. Western blot analysis confirmed the recognition of both bands by anti-bovine IFN- antibodies, suggesting the presence of two distinct isoforms of brIFN-τ (Fig. 3). Protein expression after 72 h of induction varied significantly depending on the induction strategy employed: 41 μg/mL for strategy 1, 55.54 μg/mL for strategy 2, 58.55 μg/mL for strategy 3, 72.28 μg/mL for strategy 4, and 30.03 μg/mL for strategy 5. At 120 hours (Fig. 3C-D), the measured protein yield was 97.35 μg/mL for strategy 3 and 105.05 μg/mL for strategy 4 (Fig. 3E). This latter consistently yielded higher brIFN-τ concentration than strategy 3 across the time course (Fig. 3E, blue line vs. red line). 3.2.2. Purification of brIFN-τ The culture supernatant obtained from Strategy 4 was subjected to chromatographic purification process. The first step showed a protein fraction with 80.1% purity and the elution profile a single major peak corresponding to the target protein (Fig. 4A, E1). The second step generated three distinct elution peaks (Fig. 4B, E1, E2, and E3), with brIFN-τ present only in the first two elution peaks (Fig. 4C). The final purity reached 90.1%, and the overall recovery yield of brIFN-τ was 45.63%. 3.2.3. Cell viability, antiviral markers, and antiviral activity of brIFN-τ The cell viability assay showed that purified brIFN-τ maintained 100% cell survival up to a 1:128 dilution after 24 h of exposure, indicating complete protection against virus-induced cytopathic effects. In contrast, human IFNα-2b (hIFNα-2b), used as a positive control, reduced cell survival starting at the 1:32 dilution, with lower viability observed at higher dilutions (Fig. 5, orange bars). Both brIFN-τ and hIFNα-2b treatments resulted in significantly higher cell viability (p < 0.05) compared to the negative control. The cell viability assays performed in MDBK cells exposed to the Mengo virus confirmed the consistent protective activity of brIFN-τ over hIFNα-2b across all tested dilutions (Fig. 5C). The specific antiviral activity of brIFN-τ was approximately 9.7 × 10⁹ IU/mg (Fig. 5D). No statistically significant differences were observed between the brIFN-τ doses tested, whereas all treated groups showed significantly higher expression (p < 0.05) than the untreated control group. 3.3.1. Microparticles obtained by spray drying procedure During the microencapsulation procedure of brIFN-τ, the total yield exceeded 56% (Table 6). The results from spray drying the empty chitosan control samples indicated that the encapsulation yields at different temperatures were similar (Table 7). Shape and size of the empty chitosan particles and the chitosan particles encapsulating brIFN-τ are reported in Fig. 6 A-F. The control empty chitosan particles were irrespective of the temperature conditions and remained relatively consistent (Fig. 6A, 629 C, and E). In contrast, adding brIFN-τ resulted in more irregular particle morphology and increased surface heterogeneity, attributed to the presence of the recombinant protein (Fig. 6 B, D, and F). 3.3.2. Antiviral activity of encapsulated brIFN-τ in vitro Significant differences (p<0.05) in cell survival were observed between treated groups and the negative control. Western blot analysis (Fig. 6H) showed that brIFN-τ was present in supernatants collected after 48 h of incubation, with higher released levels (p < ??) observed for particles generated at 140 °C and 120 °C, relative to those from 100 °C (Fig. 6H, lanes 1 to 3). 3.3.3. brIFN-τ release in vitro, simulating bovine uterine environment conditions The results indicate that brIFN-τ can be detected in the supernatant as early as day 1, reaching concentrations close to 100 μg/mL (Fig. 7A). However, the levels subsequently declined until day 22, when signal was observed in western blot analysis (Fig. 7B). 3.4.1. Stable hydrogel matrix synthesis The starch-to-chitosan ratio was set at 70:30 to balance stiffness, flexibility, and long-term stability (results not shown). Following the addition of the crosslinking agent, microparticles were introduced on day 2 of the procedure (Fig. 8). At this stage, no structural loss was observed in the hydrogel by the end of the process (Fig. 9A). 3.4.2. Hydrogel characterization The SEM results indicated that the hydrogel created with microparticles containing brIFN-τ (Fig. 9A) exhibited greater structural homogeneity than those produced with soluble brIFN-τ and empty microparticles. Case A demonstrated less heterogeneity despite all samples showing cavities within the 100 to 200 μm range (Fig. 9). In contrast, the hydrogels formed under the alternative conditions displayed greater variability among the cavities observed (Fig. 9B, C). The results showed that the hydrogels containing microparticles with 730 brIFN-τ provided 100% protection up to a dilution of 1:8. However, the protection began to decrease with a dilution of 1:16, reaching non-significant values at 1:32 (Fig. 9, white bars). In contrast, the hydrogels with unencapsulated soluble brIFN-τ exhibited lower protection, around 70% up to a 1:4 dilution, with subsequent decreases observed. For samples containing only purified brIFN-τ, protection was maintained up to 1:128 dilution (Fig. 9D, black bars). Significant differences (two-way ANOVA p<0.05; Shapiro-Wilk p> 0.10; Levene’s p>0.10) were noted between the black and white columns compared to control cells treated with buffers and those treated with supernatant from hydrogels containing empty particles (p<0.05). 3.4.3. brIFN-τ release from hydrogel in vitro Total protein values (Fig. 9E) began to be detected after 24 h, showing a steady increase until the tenth day, when the maximum release was recorded at 22.7 ± 1.39 μg/mL. After this peak, the protein release remained relatively constant until the end of the test on day 26, with values ranging from 17.8 ± 1.76 to 20.48 ± 2.36 μg/mL. The cumulative amount of protein released over the 26-day evaluation reached 214.4 μg/mL. Throughout the study, from day 5 to day 20, an 80% level of protection was measured, which remained above 70% until day 25 (Fig. 9F) (two-way ANOVA ??; Shapiro-Wilk p> 0.10; Levene’s p>0.10). 3.5.1. Drug safety in the ovine model Figure 10A reports the protocol used for drug safety. Regarding the behavioral assessment of sheep during the trial, no behavioral changes were detected in either group. The total individual scores for animals in both groups remained between 0 and 3, indicating that the sheep maintained normal, stable conditions and that the formulation did not impact their well-being. Concerning examining the genital area and secretions, all sheep exhibited redness, edema, and mucous secretion on day 0, which was associated with the onset of estrus (Fig. 10B, C). During the first week following treatment, there was no increase in these indicators, nor were any secretions present. Only variations in redness, edema, and secretions characteristic of the estrous cycle 800 were observed throughout the trial, with no significant differences identified between the trial groups. Pairwise comparisons indicated that on days 7 and 24, the control group had significantly lower rectal temperatures than the treated group (p < 0.05); however, these differences were not considered clinically relevant. Some measurements below 38°C were noted when ambient temperatures fluctuated between 2 and 3°C (Fig. 10D). Vulvar temperature data were analyzed using the same statistical approach. The assessment of vulvar temperature showed an average of 36.7°C for the treated group and 37.0°C for the control group at the start of the trial, when all females were in heat. Following the application of treatment, the average vulvar temperature fluctuated between 36.7°C and 37.9°C for both groups, with overall averages of 37.4°C for the treated group and 37.2°C for the control group, revealing no significant differences between them (p> 0.05) (Fig. 10F). The hematology analysis showed that values were within normal ranges for both groups at all assessed time points (Table 8). As total leukocytes regards, control group exhibited abnormal values (p < ??) on day 7, that returned to normal levels on day 14. 3.5.2. Anti-luteolytic activity validation in cows Figure 11A reports the protocol used for anti-luteolytic activity validation. Ultrasound data revealed that the administered device remained visible until day 22 in brIFN-τ microencapsulated with hydrogel group (G2), whereas non-encapsulated brIFN-τ plus hydrogel group (G3) showed visibility of the device only until day 11 (Fig. 11B). Progesterone concentration underscore the hormonal kinetics during the test, demonstrating higher and more sustained levels (p < ??) throughout the estrous cycle in G2 compared to the control group and notably maintaining higher levels (p < ??)than those observed in G3 (Fig. 11D). The results indicated decreased circulating progesterone levels in the control group due to the luteolytic effect (characteristic of a normal estrous cycle). Conversely, this decline was not observed in the group receiving microencapsulated brIFN-τ (G2) and only to a lesser extent (p < ??) in the group with non-encapsulated interferon (G3) (Fig. 11D). |
|
|
|
Response: Thank you for pointing this out. Agree. We have, accordingly, modified all Results and included your suggestion in the manuscript to emphasize this point. All values of p were actualized in the document, page 22-23 lines 750-778. |
|
|
|
4. Response to Comments on the Quality of English Language |
|
Point 1: The English could be improved to more clearly express the research. |
|
Response 1: Thank you for your observation. We have taken it into careful consideration. The manuscript has been revised for clarity, grammar, and academic style, with improvements implemented throughout the Abstract, Introduction, Results, and Discussion sections. Particular attention was paid to sentence structure, word choice, and consistency in terminology to enhance the overall readability and scientific precision of the text. |
|
5. Additional clarifications |
|
|

Reviewer 4 Report
Comments and Suggestions for Authors
The manuscript entitled “A novel microencapsulated bovine recombinant interferon Tau formulation for luteolysis modulation in cattle” is an interesting work written in clear understandable English. Authors produced system of encapsulated IFN which potentially may help saving pregnancies in cattle. However, I have multiple questions for the authors, therefore I select Major revision. Below I will provide my commentaries divided in two sections: Major and Minor.
Major
- You state that one of the main problems for saving pregnancies in cattle is reduction of progesterone production. To avoid this problem, you tried do design so complex system which will affect progesterone production. But why not to select more simple way – provide to the cattle directly progesterone? This approach is commonly used for treatment of infertility in humans worldwide: to provide synthetic progesterone (progestins) at the beginning of the pregnancy to support it. I am sure that in cattle it also should work.
- Line 108. Why you decided to perform intrauterine method of targeting, nor injection in the blood, for example?
- What is YP and YPG medium? What does mean these abbreviations? Who is manufacturer? Or it is homemade medium? What is composition of these mediums?
- Your work involved experiments on animals. Please, provide some reference that this work was accepted by your institutional bioethical committee. There are some experiments in which animals suffered from pain.
- Fig 10F is absent
- It will be interesting to see results with pregnancies and compare them with consumption of progesterone by cattle. I understand that it is another significant work not for this article, but you can discuss future directions in conclusion or discussion.
- Can the implant provoke uterine contractions (as many intrauterine implants affects the uterus) and therefore avoiding implantation? Can the implant disturb the fetus?
- Primers should be given in the table by the following manner:
Gene Name – Gene ID (NCBI gene, NM_....) – Forward primer – Reverse Rimer – Melting temperature – Product Size
I have tried to check the primers using Primer Blast, however for the OAS the system announced that the left primer cannot be found in template. Probably, it is due to the fact that I used inappropriate NM. Please, make the Table as I mentioned above.
- Line 219. Explain here what is positive and negative control groups.
- Why you selected for PCR only one reference? It is well known that it is better to use few reference genes from distinct groups.
- You investigated the action of IFN on MDBK cells, it is interesting, however it is a little bit out of the theme of your research. It will be more in scope to investigate the action of IFN on some uterine cells cultured in vitro to see the effects.
- You encapsulate interferon by heating at significant temperatures (100oC and more), however interferon is a protein. And heating leads to the protein denaturation. How did you checked that it is still interferon after heating? That this protein did not changed their structure?
- Fig 7B. Why the bands are so blurry? The positive and negative controls are absent in this image.
- Chapter 3.3.3. Why do you think that described conditions represent bovine uterus?
Minor
- Avoid point “.” At the end of the title and delete the word “Title”.
- Lines 55-57. “In Latin America, where beef and dairy exports are crucial components of 54 agricultural Gross Domestic Product (GDP), early pregnancy loss in cattle can translate 55 into millions of dollars in direct and indirect yearly losses, particularly in countries like 56 Brazil, Argentina, and Mexico [3].” The majority of the authors are from Chile. And what about Chile?
- Line 106 (and in other places in the text), the name of bacteria. In line 90 you mentioned another name of this bacteria “Komagataella phaffii (Pichia pastoris)”. As I know the actual name is Komagataella phaffi, therefore it is better to use the actual name.
- Figure 11C. Seeing this photo I cannot understand the scale of the implant. Please make a photo with a scale bar or a school ruler, for example.
- Line 267. The reference is duplicated.
- Line 250. What is SFB? Maybe FBS?
- Line 233. The formula of the Ct. You forgot the ^.
- Indicate the source of Mengo virus.
- As I understand implant should be installed prior to the ovulation? Focus more attention of the reader on the precise time of implant installation.
Author Response
|
3. Point-by-point response to Comments and Suggestions for Authors Major |
|
Comments 1: You state that one of the main problems for saving pregnancies in cattle is reduction of progesterone production. To avoid this problem, you tried do design so complex system which will affect progesterone production. But why not to select more simple way – provide to the cattle directly progesterone? This approach is commonly used for treatment of infertility in humans worldwide: to provide synthetic progesterone (progestins) at the beginning of the pregnancy to support it. I am sure that in cattle it also should work. |
|
Response 1: [Type your response here and mark your revisions in red] Thank you for pointing this out. We agree with this comment. Therefore, In addition to the antiluteolytic effect of interferon tau, its antiviral effect adds a positive effect that can benefit herds with low fertility rates. Furthermore, the temporary decrease in circulating progesterone in cows with early pregnancies does not increase embryonic or pregnancy losses. Still, it does result in a smaller embryo in early gestational stages (Carvalho et al., 2017). In contrast, exogenous P4 supplementation between 4 and 18 days post-artificial insemination (AI) increased the percentage of cows experiencing luteolysis 19 days post-AI (Monteiro et al., 2014), indicating an early downregulation of endometrial P4 receptors and the advancement of the luteolytic cascade. Monteiro PL Jr, Ribeiro ES, Maciel RP, Dias AL, Solé E Jr, Lima FS, Bisinotto RS, Thatcher WW, Sartori R, Santos JE. Effects of supplemental progesterone after artificial insemination on expression of interferon-stimulated genes and fertility in dairy cows. J Dairy Sci. 2014;97(8):4907-21. doi: 10.3168/jds.2013-7802. Epub 2014 Jun 7. PMID: 24913649. Carvalho PD, Consentini CC, Weaver SR, Barleta RV, Hernandez LL, Fricke PM. Temporarily decreasing progesterone after timed artificial insemination decreased expression of interferon-tau stimulated gene 15 (ISG15) in blood leukocytes, serum pregnancy-specific protein B concentrations, and embryo size in lactating Holstein cows. J Dairy Sci. 2017 Apr;100(4):3233-3242. doi: 10.3168/jds.2016-11996. Epub 2017 Feb 8. PMID: 28189320.
|
|
Comments 2: 2. Line 108. Why you decided to perform intrauterine method of targeting, nor injection in the blood, for example? |
|
Response 2: Thank you for pointing this out. Agree. I/We have, accordingly, revised to emphasize this point. The choice to use an intrauterine method for delivering the microencapsulated bovine recombinant interferon tau (brIFN-τ) formulation, instead of systemic injection, came from the formulation's characteristics and the biological context. For example, IFN-τ mainly works on the endometrium to stop luteolysis by reducing prostaglandin F2α (PGF2α) release and oxytocin receptor expression. Intrauterine administration delivers the treatment directly to the action site, maximizing local effectiveness while reducing systemic side effects. Systemic injections might result in rapid clearance, degradation, or off-target effects. In contrast, intrauterine delivery allows for a sustained release at the reproductive site. The chitosan-based microencapsulation and starch-chitosan hydrogel were designed for prolonged release, imitating the natural secretion pattern of endogenous IFN-τ during early pregnancy, specifically days 15 to 17. The intrauterine placement enables the hydrogel to break down slowly, releasing brIFN-τ over about 20 days, as shown in Figures 7 and 9E, corresponding with the crucial period for maternal recognition of pregnancy. Systemic administration would need frequent dosing due to the short half-life of IFN-τ. The hydrogel, however, provides continuous local delivery. |
|
Comments 3: What is YP and YPG medium? What does mean these abbreviations? Who is the manufacturer? Or it is homemade medium? What is composition of these mediums? |
|
Response 3: Thank you for pointing this out. Agree. We have, accordingly, modified to emphasize this point. YP (Yeast Extract 1%-Peptone 2%) medium was standardized for Saccharomyces cerevisiae cultures in fermentation and yeast genetics research during the 20th century. The YPG variant (with glycerol 2% as the carbon source) became popular for Pichia pastoris cultures in the 1980s, when this yeast began to be used as a recombinant protein expression system. The reference for using these media is found in Thermo Fischer's PichiaPink™ Expression System. We added the composition of the medium, which is mentioned in the article, along with the reference and the corresponding description in the abbreviation. The change can be found on page 4, paragraph 3, lines 168 and 169. In addition, these acronyms were included in the abbreviations section on page 30, lines 1055 and 1056.
Comments 4: Your work involved experiments on animals. Please, provide some reference that this work was accepted by your institutional bioethical committee. There are some experiments in which animals suffered from pain. Response 4: Thank you for pointing this out. Agree. We have, accordingly, adjunct the document to emphasize this point. The animal study protocol was approved by the Comité de Ética, Bioética y Bioseguridad of Universidad de Concepción (CEBB 1490-2023). Attached document at the end of this document. This number of the document can be found on page 30, paragraph 5, and lines 1025 and 1026.
Comments 5: Fig 10F is absent Response 5: Thank you for pointing this out. Agree. We have, accordingly, changed to emphasize this point. There is no Fig 10F, it is 10E instead. The change can be found on page 21, paragraph 3, lines 700 and in the Fig 10. |
|
Comments 6: It will be interesting to see results with pregnancies and compare them with consumption of progesterone by cattle. I understand that it is another significant work not for this article, but you can discuss future directions in conclusion or discussion. Response 6: Thank you for pointing this out. Agree. We have, accordingly, revised and added a future projection in Conclusion to emphasize this point. We plan future studies to compare pregnancy outcomes in cows with those using injectable progesterone and also to evaluate the study of our microencapsulated formulation used as an injectable product. This change can be found on page 28, paragraph 6, and line 1006-1013.
Comments 7: Can the implant provoke uterine contractions (as many intrauterine implants affect the uterus) and therefore avoid implantation? Can the implant disturb the fetus? Response 7: Thank you for pointing this out. Agree. We have, accordingly, explained this inquiry to emphasize this point. The implant did not induce contractions during trials in either sheep or cattle. Preliminary results from synchronized cows that received an artificial insemination (AI) combined with an interferon implant, compared to a control group that only received insemination, showed positive pregnancy outcomes. It can be concluded that the implant has no spermicidal effect and does not impact implantation (unpublished data). These findings are ongoing and will be the focus of a second article, which will include a larger number of animals and comprehensive pregnancy trials in the target species.
Comments 8: Primers should be given in the table by the following manner: Gene Name – Gene ID (NCBI gene, NM_....) – Forward primer – Reverse Rimer – Melting temperature – Product Size I have tried to check the primers using Primer Blast, however for the OAS the system announced that the left primer cannot be found in template. Probably, it is due to the fact that I used inappropriate NM. Please, make the Table as I mentioned above. Response 8: Thank you for pointing this out. Agree. We have accordingly modified and created a table to emphasize this point. We added to the text another table with the primer’s description. The OAS2 forward primer was missing two base al the 3`, which was rectified. In the revised manuscript this change can be found on page 6, table 3, and lines 250.
Comments 9: Line 219. Explain here what is positive and negative control groups. Response 9: Thank you for pointing this out. Agree. We have, accordingly, added the description to emphasize this point. We added to the text the description of the negative and positive controls in the activity study. In the revised manuscript this change can be found on page 5, paragraph 5, and lines 224-225.
Comments 10: Why you selected for PCR only one reference? It is well known that it is better to use few reference genes from distinct groups. Response 10: Thank you for pointing this out. Agree. I/We have, accordingly, changed to emphasize this point. We are unsure about your question. Are you asking about the use of reference or housekeeping genes? If that’s the case, we used beta-actin as our housekeeping gene in the assay. Beta-actin is one of the most commonly used reference genes in real-time PCR (qPCR) for normalizing gene expression levels. Other alternatives include GAPDH and 18S rRNA. We have primers for bovine beta-actin available in our laboratory, and they have produced reliable results in previous studies.
Comments 11: You investigated the action of IFN on MDBK cells, it is interesting, however it is a little bit out of the theme of your research. It will be more in scope to investigate the action of IFN on some uterine cells cultured in vitro to see the effects. Response 11: Thank you for pointing this out. Agree. We have, accordingly, explained to emphasize this point. In this case, because IFN-tau is a type 1 interferon, we used MDBK cells (bovine cell culture) to determine the interferon activity and its antiviral activity and thus monitor whether the interferon was active at the different stages of the process of obtaining, purifying, and subsequently encapsulating it, etc. We presented this strain in our laboratory and were not measuring the effect of pregnancy. However, a specific cell culture from the bovine uterus could be considered for future work.
Comments 12: You encapsulate interferon by heating at significant temperatures (100oC and more), however interferon is a protein. And heating leads to the protein denaturation. How did you checked that it is still interferon after heating? That this protein did not changed their structure? Response 12: Thank you for pointing this out. Agree. I/We have, accordingly, changed to emphasize this point. Interferon encapsulation was carried out at three different temperatures to assess encapsulation yield and corresponding activity assays. This study aimed to evaluate the effect of temperature on the protein's activity. Results from the cytopathic effect inhibition assay conducted in MDBK cells infected with the Mengo virus indicated that the protein retains its activity, as evidenced by the preservation of live cells when examining dilutions of the formulation supernatant. Additionally, the retention time at the chosen temperatures is approximately 1 second; however, it decreases at temperatures between 60 and 80°C in the aspiration cyclone, where it also remains for about another second before reaching the collection vessel at temperatures ranging from 25 to 30 degrees Celsius. To further monitor protein degradation, SDS-PAGE and Western blot were performed. Circular dichroism of the samples could be performed at each step to determine whether the interferon had lost structure. This study did not use this technique since the samples maintained antiviral activity at each stage.
Comments 13: Fig 7B. Why the bands are so blurry? The positive and negative controls are absent in this image. Response 13: Thank you for pointing this out. Agree. We have, accordingly, emphasized this point. The appearance of blurred bands in the release assay may be due, among other factors, to the formation of aggregates between chitosan and the protein. Chitosan (positively charged) can form complexes with proteins (negatively charged), generating aggregates that migrate heterogeneously on SDS-PAGE, producing diffuse bands or multiple bands nearby during electrophoresis, mainly if not enough reducing agent is used. Their disappearance over time is due to the depletion of the IFN encapsulated in the sample. For this reason, on day 1, release is greater, primarily due to the IFN being more exposed to the solvent, etc. The absence of controls in this assay is because, previously, assays with negative controls and the same antibody were performed (Figure 3A and Figure 6H), and we decided to take advantage of this opportunity to monitor the most significant number of samples in a single assay.
Comments 14: Chapter 3.3.3. Why do you think that the conditions described represent bovine uterus? Response 14: Thank you for pointing this out. Agree. We have, accordingly, changed to emphasize this point. A typing error was set to 120, and the test was performed at 38 degrees. This change can be found in the revised manuscript on page 18, paragraph 3, and line 631.
Minor Comments 1: Avoid point “.” At the end of the title and delete the word “Title”. Response 1: Thank you for pointing this out. Agree. We have, accordingly, done so to emphasize this point. This change can be found in the revised manuscript on page 1, lines 2 and 3.
Comments 2: Lines 55-57. “In Latin America, where beef and dairy exports are crucial components of 54 agricultural Gross Domestic Product (GDP), early pregnancy loss in cattle can translate 55 into millions of dollars in direct and indirect yearly losses, particularly in countries like 56 Brazil, Argentina, and Mexico [3].” The majority of the authors are from Chile. And what about Chile? Response 2: Thank you for pointing this out, we completely agree. We have, accordingly, changed to emphasize this point, adding the following sentence: “In Chile, abortion rates are overall underestimated, with reported cumulative incidences around 12.2% and associated economic losses estimated at USD 143 per case [4,5]” The modification can be found on page 2, paragraph 1 and lines 57-59. The corresponding references have been added in the correct format and can be found on page 39 and lines 1405-1409 as follows. 4. Gädicke, P.; Monti, G. Factors Related to the Level of Occurrence of Bovine Abortion in Chilean Dairy Herds. Prev Vet Med 2013, 110, 183–189, doi:10.1016/J.PREVETMED.2012.11.022. 5. Gädicke L’Huissier, P.; Chihuailaf, R.; Letelier, R.; Allende, R.; Ruiz, A.; Junod, T.; Gädicke L’Huissier, P.; Chihuailaf, R.; Letelier, R.; Allende, R.; et al. Monitoring Different Causal Patterns of Bovine Abortion Syndrome. Veterinaria México OA 2022, 9, doi:10.22201/FMVZ.24486760E.2022.652.
Comments 3: Line 106 (and in other places in the text), the name of bacteria. In line 90 you mentioned another name of this bacteria “Komagataella phaffii (Pichia pastoris)”. As I know the actual name is Komagataella phaffi, therefore it is better to use the actual name. Response 3: Thank you for pointing this out. Agree. We have, accordingly, explained to emphasize this point. The current name of the yeast (not bacteria) used in this study is Komagataella phaffii, but this is a new name, and the old name, Pichia pastoris, continues to be accepted due, among other things, to the fact that all the consulted bibliography refers to the previous name, which is why we declared the current name at the beginning and in parentheses the old name with which we continue to refer to the yeast.
Comments 4: Figure 11C. Seeing this photo I cannot understand the scale of the implant. Please make a photo with a scale bar or a school ruler, for example. Response 4: Thank you for pointing this out. Agree. We have, accordingly, changed and added a photo to Fig. 11C to emphasize this point. This change can be found in the revised manuscript on page 23, Fig 11C, paragraph 1 lines 746 where described the photo (4 cm length)(0.3 cm of diameter).
Comments 5: Line 267. The reference is duplicated. Response 5: Thank you for pointing this out. Agree. I/We have, accordingly, eliminated the duplicate reference to emphasize this point.
Comments 6: Line 250. What is SFB? Maybe FBS? Response 6: Thank you for pointing this out. Agree. We have, accordingly, changed to FBS to emphasize this point This change can be found on page 6, paragraph 2, and line 253-255.
Comments 7: Line 233. The formula of the Ct. You forgot the ^. Response 7: Thank you for pointing this out. Agree. We have, accordingly, corrected the formula to emphasize this point. This change can be found on page 6, paragraph 1, and line 241 .
Comments 8: Indicate the source of Mengo virus. Response 8: Thank you for pointing this out. Agree. I/We have, accordingly, changed to emphasize this point. The virus was donated from the Center for Genetic Engineering and Biotechnology, CIGB, Cuba). This change can be found on page 5, paragraph 5, and line 218-219.
Comments 9: As I understand implant should be installed prior to the ovulation? Focus more attention of the reader on the precise time of implant installation. Response 9: Thank you for pointing this out. Agree. We have, accordingly, explained this fact to emphasize this point. The implant is administered during heat, along with artificial insemination (AI), 30 seconds after AI.
The quality of the English writing is fine and does not requeri ay improvement 4. Response to Comments on the Quality of English Language |
|
Point 1: |
|
Response 1: Thank you for your observation. |
|
5. Additional clarifications |

Reviewer 5 Report
Comments and Suggestions for Authors
Dear authors,
I will recommend your article for publication, but I have several questions about the research results. They are short, but very serious to be asked.
L72 to L86 – Too much information about PGF2α, not related to aim of study.
L103 – Not find in [21] any information about IFN.
L106 – Not clear – in [16] you were declared IFN biotech production in Pichia pastoris, but in firs aim presents express and purify biologically active brIFN in Pichia pastoris. What is it?
L106 – “Biodegradable carriers such as chitosan and poly lactic-co- glycolic acid (PLGA) enhance protein stability, prolong systemic residence time and reduce dosing frequency [17–19]” and second aim is “formulate and characterize chitosan-based microencapsulation systems”. Repeating of research?
L111 – While only in cattle? This IFN universal for all ruminants.
L118 – And from where amino acid sequence of brIFN was downloaded?
L189 – Add in title where methanol concentration was established.
L192 – What and how much was added?
L192 – centrifuged before supernatant collection?
Regards,
Author Response
|
3. Point-by-point response to Comments and Suggestions for Authors Major |
|
Comments 1: L72 to L86 – Too much information about PGF2α, not related to aim of study. |
|
Response 1: Thank you for pointing this out. Agree. I/We have, accordingly modified the introduction to emphasize this point. We deleted the unnecessary information about PGF2alfa and, instead, focused on including more details about maternal recognition of pregnancy, wih a focus on the molecular aspects by rewriting that paragraph. The updated text can be found on page 2, paragraph 2, between lines 63-89
Comments 2: L103 – Not find in [21] any information about IFN. |
|
Response 2: Thank you for pointing this out. Agree. We have, accordingly, revised to emphasize this point by switching said reference for one more relevant to IFNtau release pattern. The previously [21] reference in the manuscript, talks about different uses of hydrogel formulation in the context of long-acting drug development, since that characteristic of hydrogel its key to the research we included it in the introduction. However, as you indicated, the best reference for L103 should have more relevance to IFN-t release pattern, and we modified it accordingly adding as a reference a paper by Kowalcyk that discusses IFNtau release patterns. The modification can be found on page 3, paragraph 1 and line 105.
|
|
Comments 3: L106 – Not clear – in [16] you were declared IFN biotech production in Pichia pastoris, but in firs aim presents express and purify biologically active brIFN in Pichia pastoris. What is it? |
|
Response 3: Thank you for bringing this to our attention. We have, accordingly, explained what we declared to emphasize this point. In this section, we aim to demonstrate that the Pichia pastoris expression system has been effectively used to produce and correctly fold cytokines, including interferons. This evidence supports the use of this system for expressing recombinant interferon tau with biological activity. Page 2, paragraph 3, line 93.
Comments 4: L106 – “Biodegradable carriers such as chitosan and poly lactic-co- glycolic acid (PLGA) enhance protein stability, prolong systemic residence time and reduce dosing frequency [17–19]” and second aim is “formulate and characterize chitosan-based microencapsulation systems”. Repeating of research? Response 4: Thank you for pointing this out. Agree. We have, accordingly, explained this document to emphasize this point as follows: Said papers were used as theoretical basis that biodegradable carriers possess characteristics that are attractive for application uses related to drug-release, such as prolong systemic residence time, and it also includes an updated summary of encapsulation forms of IFN-α, IFN-ß, and IFN-γ; a review that highlights the innovation required in this field. On the other hand, in this project we focused on formulating a biodegradable chitosan-based carrier for our specific interferon of interest (IFN-tau) in the very specific context of pregnancy recognition in cattle. So we use previously discussed methodology in a brand new microencapsulation system.
Comments 5: L111 – While only in cattle? This IFN universal for all ruminants. Response 5: Thank you for pointing this out. We have, accordingly, explained this point. While tau interferons (IFN-tau) can be used in ruminants, notable differences exist among those derived from bovine, ovine, and caprine sources. Although they exhibit a high degree of structural and functional homology, these differences pertain to genetic variants, biological activity, and species specificity. |
|
Comments 6: L118 – And from where amino acid sequence of brIFN was downloaded?. Response 6: Thank you for pointing this out. Agree. We have, accordingly, added the download site and number in the manuscript to emphasize this point. We downloaded the sequence from Uniprot. This change can be found on page 3, paragraph 4 and line 122. Also is declared on page 10, paragraph 1 and line 453.
Comments7: L189 – Add in title where methanol concentration was established. Response 7: Thank you for bringing this to our attention. Agree. We have, accordingly, added to the title of Table 2 to emphasize this point. This change can be found on page 5, paragraph 1, line 190.
Comments 8: L192 – What and how much was added? Response 8: Thank you for pointing this out. Agree. We have, accordingly, changed to emphasize this point. The addition of trace amounts of vitamins to the Pichia culture mentioned in the article was conducted following the methods outlined by Pedroso-Santana et al. (2020), which is reference 53 in this manuscript. This information can be found on page 5, paragraph 2, lines 195-196.
Comments 9: L192 – centrifuged before supernatant collection? Emilio Response 9: Thank you for pointing this out. Agree. We have accordingly emphasized this point in the manuscript to highlight it. The supernatant was collected by centrifugation at 17,000g for 20 minutes. This modification can be found on page 5, paragraph 2, line 196.
The quality of the English writing is fine and does not requeri ay improvement 4. Response to Comments on the Quality of English Language |
|
Point 1: |
|
Response 1: Thanks for your observation |
|
5. Additional clarifications |
|
|

Round 2
Reviewer 1 Report
Comments and Suggestions for Authors
The revised version of the manuscript entitled “A novel microencapsulated bovine recombinant interferon Tau formulation for luteolysis modulation in cattle” has been evaluated. All necessary changes have been made, and the manuscript is now suitable for publication.
Best regards
Reviewer 4 Report
Comments and Suggestions for Authors
Can be accepted.